# Modality-Inconsistent Continual Learning of Multimodal Large Language Models

**Weiguo Pian**[1]   **Shijian Deng**[1]   **Shentong Mo**[2]   **Mingrui Liu**[3]   **Yunhui Guo**[1]   **Yapeng Tian**[1]

[1]*The University of Texas at Dallas*   [2]*Carnegie Mellon University*   [3]*George Mason University*
*{weiguo.pian,shijian.deng,yunhui.guo,yapeng.tian}@utdallas.edu   shentonm@andrew.cmu.edu   mingruil@gmu.edu*

**Reviewed on OpenReview:** *https://openreview.net/forum?id=FD8or43fBU*

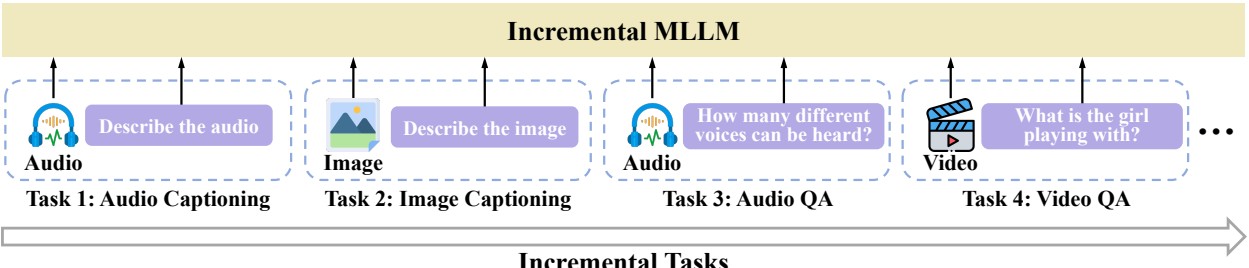

Figure 1: Illustration of our proposed Modality-Inconsistent Continual Learning (MICL), a novel and practical continual learning scenario of Multimodal Large Language Models (MLLMs), where tasks involve inconsistent modalities (image, video, or audio) and varying task types (captioning or question-answering).

## Abstract

In this paper, we introduce Modality-Inconsistent Continual Learning (MICL), a new continual learning scenario for Multimodal Large Language Models (MLLMs) that involves tasks with inconsistent modalities (image, audio, or video) and varying task types (captioning or question-answering). Unlike existing vision-only or modality-incremental settings, MICL combines modality and task type shifts, both of which drive catastrophic forgetting. To address these challenges, we propose MoInCL, which employs a Pseudo Targets Generation Module to mitigate forgetting caused by task type shifts in previously seen modalities. It also incorporates Instruction-based Knowledge Distillation to preserve the model's ability to handle previously learned modalities when new ones are introduced. We benchmark MICL using a total of six tasks and conduct experiments to validate the effectiveness of our MoInCL. The experimental results highlight the superiority of MoInCL, showing significant improvements over representative and state-of-the-art continual learning baselines.

## 1   Introduction

Multimodal Large Language Models (MLLMs), leveraging the generative capabilities of LLMs, have demonstrated remarkable performance across diverse modality-specific tasks (Li et al., 2022b; 2023; Zhang et al., 2023b; Liu et al., 2023; Panagopoulou et al., 2023; Liu et al., 2024). MLLMs typically consist of a pre-trained modality encoder, like CLIP (Radford et al., 2021) for visual data, a pre-trained LLM, and a modality adapter that projects modality-specific features into the language token space. During training, the modality encoder is usually frozen to preserve its pre-trained knowledge, while the adapter and, optionally, the LLM are fine-tuned to align cross-modal representations and enhance task performance.

While fine-tuned MLLMs have demonstrated promising performance across various multimodal tasks, including impressive zero-shot capabilities on unseen instructions (He et al., 2023), adapting to novel tasks still requires task-specific fine-tuning. Nevertheless, existing studies (He et al., 2023; Zeng et al., 2024; Zheng et al., 2024) indicate that fine-tuning MLLMs on new tasks can lead to significant performance degradation on previously learned tasks, a phenomenon known as *catastrophic forgetting*, which remains the key challenge in continual learning. To address this issue, several works explore new approaches to enable continual training of MLLMs while mitigating the catastrophic forgetting issue. For instance, He et al. (2023) introduce the continual instruction tuning scenario for multimodal large language models, and propose an adapter-based method to handle it. Zheng et al. (2024) further explore the negative forward transfer problem in continual instruction tuning of MLLMs and propose a prompt-based method to mitigate these problems. Cao et al. (2024) propose a MLLM-based continual learning framework but mainly focusing on class-incremental image classification. While existing methods have demonstrated their abilities in alleviating the catastrophic problem in the continual learning scenario of MLLMs, they primarily focus on image modality, ignoring more general multimodal scenarios beyond image. Recently, Yu et al. (2024) introduced a modality-incremental setting for MLLMs, but treated each modality as a single, non-incremental task, ignoring the incremental nature of task types within modalities.

To address these issues, in this paper, we introduce Modality-Inconsistent Continual Learning (MICL), a novel continual learning scenario for MLLMs. In MICL, different task types, such as captioning and question-answering (QA), are introduced incrementally across learning steps incorporated with inconsistent modalities, as illustrated in Fig. 1. Unlike existing incremental learning settings of MLLMs, MICL not only highlights the modality-inconsistent (modality-incremental) scenario but also emphasizes the potential catastrophic forgetting problem arising from task type incrementality combined with modality inconsistency.

Moreover, we propose MoInCL (**Mo**dality-**In**consistent **C**ontinual **L**earning), a novel continual learning approach designed to address the MICL problem. By leveraging the generative capabilities of the LLM backbone, MoInCL introduces a *Pseudo Target Generation Module (PTGM)* to handle the task type shifts inherent in the task. Additionally, an *Instruction-based Knowledge Distillation (IKD)* constraint for LLM backbone is incorporated to preserve its ability to understand modality- and task-aware knowledge, preventing the degradation of its learned capabilities.

We evaluate our method across image, audio, and video modalities, combined with captioning and question-answering (QA) tasks, resulting in six multimodal incremental tasks (Image Captioning, Image QA, Audio Captioning, Audio QA, Video Captioning, and Video QA). Our experiments demonstrate that MoInCL significantly outperforms representative and state-of-the-art continual learning methods, effectively addressing both modality and task type shifts within MICL. In summary, this paper contributes the following:

- We propose the Modality-Inconsistent Continual Learning, a more general and practical continual learning scenario of MLLMs, where different modalities are introduced incrementally combined with different task types.

- We propose a novel continual learning approach named MoInCL to tackle the task. In MoInCL, a *Pseudo Target Generation Module (PTGM)* is introduced to address the task type shift problem of previously learned modalities through incremental steps. Moreover, we propose the *Instruction-based Knowledge Distillation (IKD)* constraint to prevent the LLM from the forgetting of learned both modality- and task-aware knowledge in old tasks.

- We benchmark the proposed MICL across three modalities—image, audio, and video—and two task types: captioning and question-answering, resulting in six incremental tasks. Experimental results demonstrate that our approach, MoInCL, significantly outperforms representative and state-of-the-art continual learning methods, showcasing its effectiveness in mitigating catastrophic forgetting from both modality and task type perspectives.

## 2 Related Work

### 2.1 Multimodal Large Language Models

Recent advances have extended Large Language Models (LLMs) to handle multimodal inputs such as images, audio, and video. Early work like CLIP (Radford et al., 2021) demonstrated the effectiveness of aligning textual and visual representations for zero-shot image classification. Flamingo (Alayrac et al., 2022) further integrated vision encoders with LLMs via cross-attention, significantly improving visual question answering (VQA) and image captioning. Subsequent models like BLIP (Li et al., 2022b) and PaLM-E (Driess et al., 2023) scaled multimodal pre-training, with BLIP using a two-stage training strategy and PaLM-E incorporating embodied reasoning. More recently, LLaVA (Liu et al., 2023), InstructBLIP (Dai et al., 2023), X-InstructBLIP (Panagopoulou et al., 2023), Audio Flamingo (Kong et al., 2024; Ghosh et al., 2025; Goel et al., 2025), VideoLLaMA (Zhang et al., 2023a; Cheng et al., 2024; Zhang et al., 2025), Qwen-VL (Wang et al., 2024; Bai et al., 2025b), etc., have leveraged instruction tuning to refine the alignment between multimodal inputs and language, pushing the boundaries of multimodal reasoning and generation. Despite this progress, challenges persist as models scale to new modalities or tasks. Effectively integrating each modality without degrading performance on others remains a key issue. Moreover, robust continual learning strategies are crucial to prevent catastrophic forgetting and maintain knowledge across both previously learned and newly introduced modalities as new modalities or task types are integrated.

### 2.2 Continual Learning

Continual learning aims to enable models to learn incrementally while retaining previously acquired knowledge. Regularization-based methods, such as Elastic Weight Consolidation (EWC) (Kirkpatrick et al., 2017), assign importance to model parameters to prevent drastic updates (Kim et al., 2023). Knowledge distillation (KD) (Li & Hoiem, 2017; Rebuffi et al., 2017; Pian et al., 2023; Mo et al., 2023; Ahn et al., 2021; Douillard et al., 2020; Sun et al., 2025) and memory replay (Rebuffi et al., 2017; Pian et al., 2024; Chaudhry et al., 2019; Lopez-Paz & Ranzato, 2017) are other common strategies, where KD-based methods preserve past learned knowledge by aligning the predictions or internal features of a new model with those of an older one, and memory replay-based methods utilize a small memory set to store samples from old tasks, allowing the model to review a small number of old data while training on the current task (Rebuffi et al., 2017; Pian et al., 2024; Chaudhry et al., 2019; Lopez-Paz & Ranzato, 2017). Pseudo-rehearsal approaches (Odena et al., 2017; Ostapenko et al., 2019) take this a step further by generating synthetic examples via a generative model, reducing the need to store large amounts of data.

For MLLMs, where multiple modalities (e.g., images, audio, video) interact with language models, catastrophic forgetting is especially severe. Recent adapter-based continual instruction tuning (He et al., 2023) and prompt-based strategies (Zheng et al., 2024) help retain previously learned knowledge. HiDe-LLaVA (Guo et al., 2025) proposes a hierarchical decoupling strategy to separate instruction and perception components, allowing better task adaptation. SEFE (Chen et al., 2025) addresses forgetting by distinguishing between essential and superficial knowledge in continual instruction tuning. CL-MoE (Huai et al., 2025) introduces a dual momentum mixture-of-experts framework for continual visual question answering. However, these approaches mainly target image-text modalities. A modality-incremental scenario (Yu et al., 2024) has been explored, treating each modality as a separate task. However, it does not fully address evolving task types within each modality. To tackle this gap, we propose a new Modality-Inconsistent Continual Learning (MICL) scenario along with a novel approach to handle it effectively.

## 3 Method

### 3.1 Problem Formulation

In this subsection, we formalize the definition of our proposed Modality-Inconsistent Continual Learning (MICL). Given a sequence of $T$ tasks $\{\mathcal{T}_1, \mathcal{T}_2, \ldots, \mathcal{T}_T\}$, MICL aims to train the Multimodal Large Language Model (MLLM) $\mathcal{F}_{\boldsymbol{\Theta}}$ with parameters $\boldsymbol{\Theta}$ across these tasks incrementally. For the $i$-th task $\mathcal{T}_i$, we have

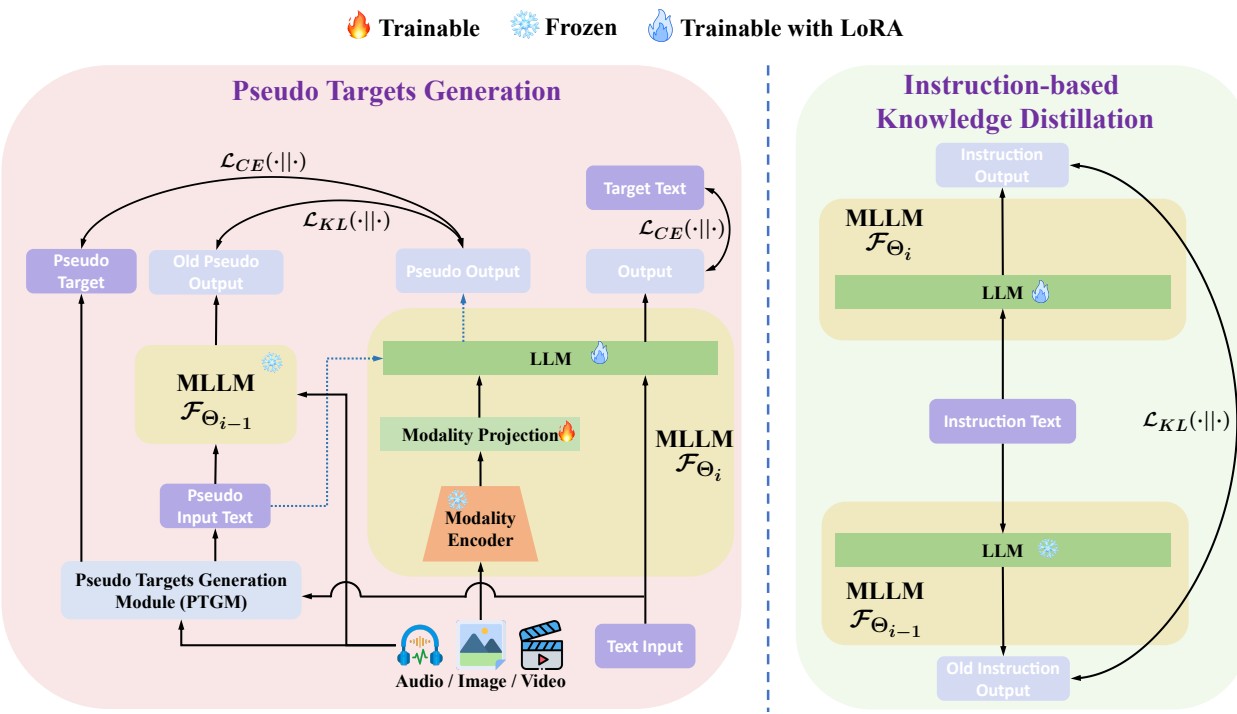

Figure 2: Overview of our proposed MoInCL, which mainly consists of a Multimodal Large Language Model (MLLM), a Pseudo Target Generation Module (PTGM), and a Instruction-based Knowledge Distillation (IKD). The red fire icon denotes the component is trainable in the current task, and the snowflake icon denotes the component is frozen during the training of the current task, while the blue fire icon means the associate component is trainable with LoRA (Hu et al., 2022) when training on the current task.

$\mathcal{T}_i = \{(\boldsymbol{x}_{i,j}, \boldsymbol{t}_{i,j}, \boldsymbol{y}_{i,j})_{j=1}^{n_i}, M_i, P_i\}$, where $M_i$ and $P_i$ denote the modality and task type of task $\mathcal{T}_i$, respectively. $\boldsymbol{x}_{i,j}$, $\boldsymbol{t}_{i,j}$, and $\boldsymbol{y}_{i,j}$ present the modality's input data, the input text, and the target text of the $j$-th data sample of task $\mathcal{T}_i$. In our setting, the input text $\boldsymbol{t}_{i,j}$ varies depending on the task type. For captioning tasks, it may consist of a simple instruction, such as "`Describe the image/video/audio.`" For question-answering (QA) tasks, the input text consists of sample-specific questions tailored to each instance. Moreover, the target text $\boldsymbol{y}_{i,j}$ typically consists of detailed description sentences for captioning tasks, while for QA tasks, it is usually limited to a few answer words. *Please note that, the output $\boldsymbol{y}_{i,j}$ is always a text sequence, consistent with the design of LLMs and MLLMs, which generate natural language outputs across diverse tasks. Tasks with non-textual outputs (e.g., image or video generation) are beyond the scope of our current formulation, as they typically require fundamentally different architectures and objectives.* We define $\mathcal{D}_i = \{(\boldsymbol{x}_{i,j}, \boldsymbol{t}_{i,j}, \boldsymbol{y}_{i,j})_{j=1}^{n_i}\}$ as the available training data when training the model $\mathcal{F}_{\boldsymbol{\Theta}}$ on task $\mathcal{T}_i$. Following the settings in modality-incremental learning (Yu et al., 2024), we do not include the memory set for replay in our MICL scenario, resulting in a memory-free continual learning setting. Please note that the term memory-free refers strictly to the absence of replay buffers that store multimodal samples from previous tasks. In our proposed MoInCL method, we introduce an auxiliary text-only instruction corpus solely for the IKD component (see Sec. 3.4). This corpus is not part of the MICL task stream, contains no modality-specific data, and it serves only as a regularization resource within our proposed MoInCL and does not alter the underlying memory-free problem formulation. In summary, the training process on an incremental task $\mathcal{T}_i$ can be presented as:

$$\boldsymbol{\Theta}_i = \underset{\boldsymbol{\Theta}_{i-1}}{\arg\min} \, \mathbb{E}_{(\boldsymbol{x}, \boldsymbol{t}, \boldsymbol{y}) \sim \mathcal{D}_i} [\mathcal{L}(\mathcal{F}_{\boldsymbol{\Theta}_{i-1}}(\boldsymbol{x}, \boldsymbol{t}), \boldsymbol{y})], \quad (1)$$

where $\mathcal{L}$ denotes the cross-entropy loss function between the generated results and the target text for training the MLLM.

Please note that, in this work, we focus on two task types (captioning and question-answering) since they are among the most commonly studied in multimodal and continual learning scenarios (He et al., 2023; Yu et al., 2024). Following common practice, we adopted these tasks to establish benchmarks for comparison. Additionally, most of the multimodal tasks, such as audio-visual event localization (Tian et al., 2018; Wu et al., 2019), vision-language navigation (Zhu et al., 2020; Song et al., 2025), etc, can often be reformulated into question-answering tasks, making these two task types a natural choice in our setting.

## 3.2 Framework Overview

To address our proposed Modality-Inconsistent Continual Learning (MICL), we introduce a novel continual learning method, **MoInCL**, as illustrated in Fig. 2. MoInCL primarily comprises a Pseudo Target Generation Module (PTGM) and an Instruction-based Knowledge Distillation (IKD) constraint. For the MLLM, we adopt the commonly used architecture, containing a modality encoder, a projection layer (MLP), and an LLM, following the design paradigm used in models such as LLaVA (Liu et al., 2023), Qwen-VL family (Wang et al., 2024; Bai et al., 2025b;a), etc. However, we do not directly use the pre-trained parameters from these models, as most are designed to process only the visual modality, and their vision-language pre-training may introduce biases and unfairness in our continual learning setting. Please note that, for fair comparison, all the baseline methods use the same model architecture as our method, and during training, the modality encoders remain frozen, while the LLM is fine-tuned using LoRA (Hu et al., 2022).

## 3.3 Pseudo Target Generation Module

We now describe the Pseudo Target Generation Module (PTGM). Our key motivation is to leverage the text generation capability of the LLM component in the MLLM to address the task type shift challenge in continual learning. PTGM generates input and target text for different task types based on the modality input data of the current task. By utilizing the generated pseudo input text and pseudo targets, the model can effectively handle both the current task type and previously learned task types within the current modality.

In our PTGM, we maintain a set $LM = \{\}$ to represent all learned modalities. For example, $LM = \{"image", "audio"\}$ indicates that the model has been trained on tasks involving image or audio modalities. And for learned modalities, we maintain a modality-specific set $LT_M = \{\}$ to denote the learned task types of modality $M$. For instance, $LT_{image} = \{"captioning"\}$ if only image captioning task has been learned for image modality. Since different task types have distinct forms, the pseudo target generation process varies accordingly for each task type. Specifically, for a current task $\mathcal{T}_i$ with the modality of $M_i$, if $M_i$ is a learned modality, *i.e.* $M_i \in LM$, the PTGM will be used to generate pseudo targets for task types within $LT_{M_i}$. If $"captioning" \in LT_{M_i}$, the pseudo input text should be a simple instruction guiding the model to generate a description of the input data. In this case, the pseudo input text generation process can be implemented by automatically filling the template to produce the result "`Describe the` $M_i$". On the other hand, if $"QA" \in LT_{M_i}$, directly applying a template is not suitable, as the pseudo QA pair should be specifically tailored to the modality's data rather than relying on generic templates. To overcome this issue, we utilize the generation ability of the LLM to generate the pseudo QA pair from the caption text of the current modality's data. Please note that in our MICL scenario, the task types considered are captioning and question-answering. Therefore, when generating pseudo QA pairs, the current task should correspond to the captioning task of the current modality. To generate QA pairs from captions, we employ a three-round generation process by prompting the pre-trained LLM component of the MLLM $\mathcal{F}$. Details of this process can be found in the Appendix. In summary, we use the following formulation to denote the pseudo target generation process:

$$
\begin{aligned}
\tilde{\boldsymbol{t}}, \tilde{\boldsymbol{y}} &= PTGM(\boldsymbol{x}, \boldsymbol{y}, p), \\
s.t.\ M_i &\in LM,\ P_i \notin LT_{M_i},
\end{aligned}
\tag{2}
$$

where $p \in LT_{M_i}$ is a learned task type of modality $M_i$ (please note that $p \neq P_i$), $\tilde{\boldsymbol{t}}$ and $\tilde{\boldsymbol{y}}$ denote the generated pseudo input text and pseudo target, respectively. $\boldsymbol{x}$ and $\boldsymbol{y}$ are the modality data and target text sampled from $\mathcal{D}_i$. For task $\mathcal{T}_i$ with modality $M_i$ and task type $P_i$, if $M_i$ is a previously seen modality (*i.e.*, $M_i \in LM$),

and if the current task type $P_i$ is not a learned task type of the current modality $M_i$ (*i.e.*, $P_i \notin LT_{M_i}$), we generate pseudo targets via $PTGM$ for the learned task type $p$ of modality $M_i$. Please note that only $\boldsymbol{x}$ is used for generating pseudo captions, while only $\boldsymbol{y}$ is utilized for generating pseudo QA pairs.

After obtaining the pseudo input text and pseudo target, a dual consistency constraint is applied between (1) the pseudo outputs of the current model $\mathcal{F}_{\boldsymbol{\Theta}_i}$ and the old model $\mathcal{F}_{\boldsymbol{\Theta}_{i-1}}$, and (2) the pseudo target and the pseudo output of the current model. This process is formulated as:

$$\mathcal{L}_{p.} = \mathbb{E}_{(\boldsymbol{x},\boldsymbol{t})\sim\mathcal{D}_i}\Big[\lambda_i \mathcal{L}_{CE}(\hat{\boldsymbol{y}}'||\tilde{\boldsymbol{y}}) + \lambda_i' \mathcal{L}_{KL}(\hat{\boldsymbol{y}}'||\hat{\boldsymbol{y}}_{\boldsymbol{o}}')\Big],$$
$$s.t.\ \hat{\boldsymbol{y}}' = \mathcal{F}_{\boldsymbol{\Theta}_i}(\boldsymbol{x},\tilde{\boldsymbol{t}}),\ \hat{\boldsymbol{y}}_{\boldsymbol{o}}' = \mathcal{F}_{\boldsymbol{\Theta}_{i-1}}(\boldsymbol{x},\tilde{\boldsymbol{t}}),$$

(3)

where $\hat{\boldsymbol{y}}_{\boldsymbol{o}}'$ and $\hat{\boldsymbol{y}}'$ denote the pseudo output from the old model and current model, respectively. $\lambda_i$ and $\lambda_i'$ present the weights to balance the two loss values for task $\mathcal{T}_i$.

### 3.4 Instruction-based Knowledge Distillation

In the previous subsection, we introduced the proposed PTGM to address the task type shift problem in the MICL scenario. However, when new modalities are introduced, the model faces a modality shift, leading to catastrophic forgetting of previously learned modalities. Additionally, as the PTGM generates pseudo targets only for seen modalities, the task type shift problem persists when training on tasks involving novel modalities. Furthermore, different modalities do *not* share the modality encoder or the modality projection, meaning that the shift problems primarily arise from updates to the LLM component in the MLLM. This results in the degradation of the LLM's ability to handle previously learned modalities. To address these issues, we propose Instruction-based Knowledge Distillation (IKD), a text instruction-based constraint designed to prevent the LLM from forgetting its learned capabilities in dealing with old modalities. Specifically, as illustrated in Fig. 2, IKD aligns the outputs of the LLM component from both the old and current models by applying a consistency loss, *i.e.* KL divergence, on their responses to the same text instruction input. In this way, instead of merely learning to handle tasks from new modalities, the current LLM's generative ability is also aligned with that of the previous LLM, thereby mitigating degradation in its ability to handle previously learned modalities. To achieve this, we introduce a pure text instruction set within IKD, which is maintained throughout the incremental steps. Since this pure text instruction set contains only text and no modality-specific data, it is not considered part of any multimodal tasks in our MICL scenario. As a result, maintaining this set does not violate the continual learning constraint that prohibits access to data from previous tasks during future tasks. This process can be formulated as:

$$\mathcal{L}_{ins.} = \mathbb{E}_{\boldsymbol{t}'\sim\mathcal{I}}\Big[\mathcal{L}_{KL}(f_{\boldsymbol{\theta}_i}(\boldsymbol{t}')||f_{\boldsymbol{\theta}_{i-1}}(\boldsymbol{t}'))\Big],$$

(4)

where $\mathcal{I}$ denotes the pure text instruction set, $f_{\boldsymbol{\theta}_i}$ and $f_{\boldsymbol{\theta}_{i-1}}$ denote the LLM component of the $\mathcal{F}_{\boldsymbol{\Theta}_i}$ and $\mathcal{F}_{\boldsymbol{\Theta}_{i-1}}$, respectively.

Please note that, the text instruction set used in IKD is a pure text corpus and does not contain any modality-specific inputs. In contrast, each continual task in MICL is trained on modality-specific datasets, where supervision depends on multimodal inputs. Therefore, the instruction corpus does not overlap with the modality-specific data distributions of the continual tasks. IKD regularizes the shared LLM representations that are reused across different modalities. Since modality shift arises from updating this shared backbone when adapting to new modality-conditioned tasks, stabilizing the shared LLM helps mitigate cross-modality interference rather than exploiting any domain overlap in the instruction corpus.

### 3.5 Overall Training Target

Above, we present our proposed Pseudo Target Generation Module (PTGM) and Instruction-based Knowledge Distillation (IKD) constraint. When training on a current task $\mathcal{T}_i$, we have the main loss function:

$$\mathcal{L}_{main} = \mathbb{E}_{(\boldsymbol{x},\boldsymbol{t},\boldsymbol{y})\sim\mathcal{D}_i}\Big[\mathcal{L}_{CE}(\hat{\boldsymbol{y}}||\boldsymbol{y})\Big],$$
$$s.t.\ \hat{\boldsymbol{y}} = \mathcal{F}_{\boldsymbol{\Theta}_i}(\boldsymbol{x},\boldsymbol{t}),$$

(5)

---

**Algorithm 1** Training of MoInCL on task $\mathcal{T}_i$

---

**Require:** Old model $\mathcal{F}_{\boldsymbol{\Theta}_{i-1}}$, training set $\mathcal{D}_i$, pure text instruction set $\mathcal{I}$, current modality $M_i$, current task type $P_i$, learned modalities set $LM$, learned task type for the current modality $LT_{M_i}$ (only if $M_i \in LM$), learning rate $\eta$, scalars $\lambda_i, \lambda'_i, \alpha_i$

 1: Initialize current model $\mathcal{F}_{\boldsymbol{\Theta}_i}$ from $\mathcal{F}_{\boldsymbol{\Theta}_{i-1}}$
 2: **if** $M_i \notin LM$ **then**
 3:     $\{\} \to LT_{M_i}$
 4: **end if**
 5: **while** not converged **do**
 6:     Sample data $(\boldsymbol{x}, \boldsymbol{t}, \boldsymbol{y}) \sim \mathcal{D}_i$
 7:     $\mathcal{L} = \mathcal{L}_{CE}(\mathcal{F}_{\boldsymbol{\Theta}_i}(\boldsymbol{x}, \boldsymbol{t}) \| \boldsymbol{y})$
 8:     **if** $M_i \in LM$ and $LT_{M_i} \neq \varnothing$ **then**
 9:         $\tilde{\boldsymbol{t}}, \tilde{\boldsymbol{y}} = PTGM(\boldsymbol{x}, \boldsymbol{y}, p),\ s.t.\ p \in LT_{M_i}$
10:         $\hat{\boldsymbol{y}}' = \mathcal{F}_{\boldsymbol{\Theta}_i}(\boldsymbol{x}, \tilde{\boldsymbol{t}}), \hat{\boldsymbol{y}}'_o = \mathcal{F}_{\boldsymbol{\Theta}_{i-1}}(\boldsymbol{x}, \tilde{\boldsymbol{t}})$
11:         $\mathcal{L}_{p.} = \lambda_i \mathcal{L}_{CE}(\hat{\boldsymbol{y}}' \| \tilde{\boldsymbol{y}}) + \lambda'_i \mathcal{L}_{KL}(\hat{\boldsymbol{y}}' \| \hat{\boldsymbol{y}}'_o)$
12:         $\mathcal{L} = \mathcal{L} + \mathcal{L}_{p.}$
13:     **end if**
14:     Sample instruction data $\boldsymbol{t}' \sim \mathcal{I}$
15:     $\mathcal{L}_{ins.} = \mathcal{L}_{KL}(f_{\boldsymbol{\theta}_i}(\boldsymbol{t}') \| f_{\boldsymbol{\theta}_{i-1}}(\boldsymbol{t}'))$
16:     $\mathcal{L} = \mathcal{L} + \mathcal{L}_{ins.}$
17:     $\boldsymbol{\Theta}_i \leftarrow \boldsymbol{\Theta}_i - \eta \nabla \mathcal{L}$
18:     $\boldsymbol{\theta}_i \leftarrow \alpha_i \boldsymbol{\theta}_i + (1 - \alpha_i)\boldsymbol{\theta}_{i-1}$
19: **end while**

---

where $\hat{\boldsymbol{y}}$ is the output of the output of the current model $\mathcal{F}_{\boldsymbol{\Theta}_i}$ by taking data samples from current task's training data $\mathcal{D}_i$ as input.

Finally, in our overall training target, the dual consistency constraint for generated pseudo targets $\mathcal{L}_{pseudo}$ and the IKD constraint $\mathcal{L}_{ins.}$ are combined with the main training loss of task $\mathcal{T}_i$:

$$\mathcal{L} = \mathcal{L}_{main} + \mathcal{L}_{p.} + \mathcal{L}_{ins.} \tag{6}$$

Additionally, inspired by the parameters/weights fusion mechanism proposed in existing works (Xiao et al., 2023; Sun et al., 2024), which have demonstrated effectiveness in preserving learned knowledge from previous tasks by applying a weighted sum between the old and current models' parameters/weights, we also adopt the parameters fusion mechanism on the LLM component of the MLLM to further prevent it from forgetting the capabilities of handling previously learned modalities, which can be denoted as:

$$\boldsymbol{\theta}_i = \alpha_i \boldsymbol{\theta}_i + (1 - \alpha_i)\boldsymbol{\theta}_{i-1}, \tag{7}$$

where $\boldsymbol{\theta}$ denotes the parameters of the LLM component of the MLLM, $\alpha_i$ is the weight for balancing the two groups of parameters. The overall algorithm of our proposed MoInCL is presented in Alg. 1.

### 3.6 Distinction from Existing Methods

Our MoInCL introduces two key innovations: 1) a Pseudo Target Generation Module (PTGM) to leverage the text generation capability of the LLM component in the MLLM to address the task type shift challenge in our proposed MICL scenario, and 2) an Instruction-based Knowledge Distillation (IKD) constraint to tackle the modality shift problem in the LLM component of the MLLM.

While existing works also utilize knowledge distillation techniques to preserve knowledge from old tasks, they primarily focus on distilling final outputs or internal features between old and current models by taking same training samples as input, as seen in methods like LwF (Li & Hoiem, 2017) and EWF (Xiao et al., 2023). These approaches do not perform well in our MICL scenario, as they significantly constrain the

MLLM's ability to learn new tasks, particularly in settings with substantial gaps between tasks, such as in our proposed MICL. In contrast, our IKD leverages pure text instructions as the input to the LLM component for knowledge distillation, avoiding introducing negative impacts on the current training task. This approach allows us to directly distill knowledge of the LLM without imposing additional constraints on the MLLM's ability to learn new tasks, ensuring that both knowledge preservation and new task learning are achieved effectively in MICL.

As for the weight fusion strategy, we acknowledge that it is not one of our primary technical contributions. However, our experiments demonstrate that this strategy can be seamlessly integrated with PTGM and IKD to further enhance the anti-forgetting capability of our approach. For this reason, we also include the weight fusion strategy in our method.

## 4 Experiments

### 4.1 Experimental Setup

#### 4.1.1 Dataset

In our proposed Modality-Inconsistent Continual Learning (MICL), we include six tasks: Image Captioning, Image QA, Audio Captioning, Audio QA, Video Captioning, and Video QA. Each task is represented by a commonly used dataset. Specifically, we use the Flickr30K (Young et al., 2014) dataset for the Image Captioning task, the OK-VQA (Marino et al., 2019) dataset for the Image QA task, the AudioCaps (Kim et al., 2019) dataset for the Audio Captioning task, the Clotho-AQA (Lipping et al., 2022) dataset for the Audio QA task, the MSR-VTT (Xu et al., 2016) dataset for the Video Captioning task, and the MSVD-QA (Xu et al., 2017) dataset for the Video QA task. More dataset details are provided in the Appendix B.

#### 4.1.2 Baselines

In our experiments, we compare our proposed MoInCL with the following continual learning methods: Fine-tuning, LwF (Li & Hoiem, 2017), EWC (Kirkpatrick et al., 2017), EWF (Xiao et al., 2023), PathWeave (Yu et al., 2024), BECAME (Li et al., 2025), and HiDe-LLaVA (Guo et al., 2025). Among these, LwF, EWC, EWF, and BECAME are representative general continual learning methods, while PathWeave is the most recent state-of-the-art continual learning method designed for MLLMs, which involves a modality-aware adapter-in-adapter mechanism to address the modality-shift problem in modality-incremental learning of MLLMs. And HiDe-LLaVA is state-of-the-art LoRA-based continual learning method for MLLMs. Please note that, for a fair comparison, all baseline methods use the same model architecture as our approach, including the Large Language Model (LLM) component. We also conduct the experiment of joint training with all tasks as the Upper-Bound.

#### 4.1.3 Evaluation Metrics

Following Panagopoulou et al. (2023), we use the CIDEr score (Vedantam et al., 2015) and prediction accuracy as evaluation metrics to evaluate captioning tasks and QA tasks, respectively. For all baselines and our method, we report the average final performance across all learned tasks, *i.e.*, the average performance of all tasks after completing the training of the final task. Since captioning and QA tasks use different evaluation metrics, we separately report the average final performance for each task type: the average final CIDEr score for captioning tasks and the average final accuracy for QA tasks. We formulate them as:

$$Avg.CIDEr = \frac{1}{N_{cap.}} \sum_{i=1}^{T} c_i^T,$$
$$s.t. \ P_i = \text{"Captioning"},$$

(8)

where $N_{cap.}$ denotes the number of captioning tasks, $c_i^T$ denotes the CIDEr score of task $\mathcal{T}_i$ after completing the training of task $\mathcal{T}_T$ if task $\mathcal{T}_i$ is a captioning task. Similarly, the average final accuracy can be formulated

as:

$$Avg.Acc. = \frac{1}{N_{QA}} \sum_{i=1}^{T} a_i^T,$$

$$s.t. \; P_i = \text{``QA''},$$

(9)

where $N_{QA}$ denotes the number of QA tasks, $a_i^T$ denotes the accuracy of task $\mathcal{T}_i$ after completing the training of task $\mathcal{T}_T$ if task $\mathcal{T}_i$ is a QA task. Furthermore, to evaluate the anti-forgetting capability of each method, we propose two metrics: the forgetting ratio and the average forgetting ratio. The forgetting ratio measures the proportion of performance drop for each task after completing the training of the final task, while the average forgetting ratio represents the mean forgetting ratio across all tasks, which can be formulated as:

$$Forget._i = (s_i^i - s_i^T)/s_i^i,$$

$$Avg.Forget. = \frac{1}{T} \sum_{i=1}^{T} Forget._i$$

(10)

where $s_i^i$ and $s_i^T$ denotes the testing score of task $\mathcal{T}_i$ after the training of task $\mathcal{T}_i$ and $\mathcal{T}_T$, respectively.

Table 1: Results on the two task orders for different methods. Bold values indicate the best results in each column, while underlined values represent the second-best results in each column.

| Methods | Order 1 | | | Order 2 | | |
|---|---|---|---|---|---|---|
| | Avg. CIDEr ↑ | Avg. Acc. ↑ | Avg. Forget. ↓ | Avg. CIDEr ↑ | Avg. Acc. ↑ | Avg. Forget. ↓ |
| Fine-tuning | 30.64 | 40.58 | 41.17% | 10.82 | 37.01 | 65.56% |
| LwF (Li & Hoiem, 2017) | 34.80 | 40.21 | 39.26% | 12.37 | 38.79 | 61.84% |
| EWC (Kirkpatrick et al., 2017) | 39.06 | 37.04 | 38.79% | 9.92 | 37.65 | 66.40% |
| EWF (Xiao et al., 2023) | 24.59 | 36.34 | 48.55% | 13.92 | 45.85 | 46.64% |
| PathWeave (Yu et al., 2024) | 34.20 | 36.19 | 44.36% | 11.11 | 41.13 | 61.47% |
| BECAME (Li et al., 2025) | 24.36 | 38.50 | 46.96% | 10.61 | 43.20 | 54.10% |
| HiDe-LLaVA (Guo et al., 2025) | 25.27 | 38.35 | 46.47% | 13.93 | **46.15** | 46.18% |
| **MoInCL (Ours)** | **55.31** | **42.29** | **14.21%** | **51.13** | 45.22 | **8.93%** |
| Upper Bound (Joint training) | 66.69 | 48.97 | - | 66.69 | 48.97 | - |

### 4.1.4 Implementation Details

We implement our experiments using Pytorch (Paszke et al., 2019) and LaVIS (Li et al., 2022a) framework. For the LLM backbone of the Multimodal Large Language Model (MLLM), we adopt the Llama-3.2-1B-Instruct (Dubey et al., 2024), which is initialized with its official pre-trained parameters at the beginning of the first task. Following the implementation in (Panagopoulou et al., 2023), we apply the EVA-CLIP-ViT-G/14 (Fang et al., 2023) as the Image Encoder and Video Encoder, and the BEATs$_{\text{iter3+}}$ (Chen et al., 2023) as the Audio Encoder. Each video input consists of 4 frames, and the audio input also consists of 4 frames with the sampling rate of 11kHz. For the video and audio modalities, the Video Encoder and Audio Encoder process each frame individually and then concatenate the encoded patches from all frames, following the approach in (Panagopoulou et al., 2023). For the Image Projection, we use a two-layers MLP with the GELU (Hendrycks & Gimpel, 2016) activation function. For the Video and Audio Projection, both of them include a single convolutional layer as a pooling layer to reduce the total number of patches, followed by a two-layers MLP with the GELU activation function. For each task, we train the model using the AdamW (Loshchilov & Hutter, 2019) optimizer with an initial learning rate of 1e-5, adjusted using the cosine decay strategy, and a weight decay of 5e-2. We train our proposed MoInCL and all baseline methods on a NVIDIA RTX A6000 Ada GPU. During the training of our approach, the pure text instructions in the Instruction-based Knowledge Distillation (IKD) constraint are randomly sampled from the Natural Instructions (Mishra et al., 2022) dataset.

### 4.2 Experimental Comparison

We conduct experiments using two random task orders. For **Order 1**, the tasks are arranged as: *Audio Captioning → Image Captioning → Video QA → Audio QA → Image QA → Video Captioning*. For **Order 2**,

the task sequence is: *Image Captioning → Video Captioning → Video QA → Image QA → Audio Captioning → Audio QA*. Additional experimental results on more task orders are provided in Appendix C, where we demonstrate that our framework can handle highly challenging task orders involving severe modality and task shifts.

Table 2: Detailed testing results of the first three tasks of Order 2. The evaluation metric used for the Flickr30K and MSR-VTT datasets is CIDEr score, while that for the MSVD-QA dataset is accuracy.

| Methods | | Flickr30k | MSR-VTT | MSVD-QA |
|---|---|---|---|---|
| Fine-tuning | Step 1 | 77.50 | - | - |
| | Step 2 | 64.04 | 48.03 | - |
| | Step 3 | 12.12 | 8.64 | 46.20 |
| LwF (Li & Hoiem, 2017) | Step 1 | 77.50 | - | - |
| | Step 2 | 53.87 | 48.70 | - |
| | Step 3 | 10.20 | 7.80 | 47.64 |
| EWC (Kirkpatrick et al., 2017) | Step 1 | 77.50 | - | - |
| | Step 2 | 62.65 | 47.73 | - |
| | Step 3 | 10.45 | 9.66 | 45.79 |
| EWF (Xiao et al., 2023) | Step 1 | 77.50 | - | - |
| | Step 2 | 69.16 | 45.30 | - |
| | Step 3 | 56.10 | 9.69 | 45.33 |
| PathWeave (Yu et al., 2024) | Step 1 | 77.22 | - | - |
| | Step 2 | 53.60 | 50.01 | - |
| | Step 3 | 7.36 | 8.35 | 47.87 |
| BECAME (Li et al., 2025) | Step 1 | 77.50 | - | - |
| | Step 2 | 77.22 | 47.64 | - |
| | Step 3 | 52.16 | 9.82 | 47.35 |
| MoInCL (Ours) | Step 1 | 77.50 | - | - |
| | Step 2 | 73.59 | 48.03 | - |
| | Step 3 | 70.88 | 48.34 | 43.11 |

Table 3: Forgetting ratio of each task in Order 1. Bold values denote the best results in each column, while underlined values indicate the second-best results in each column.

| Methods | AudioCaps | Flickr30k | MSVD-QA | Clotho-AQA | OK-VQA | MSR-VTT |
|---|---|---|---|---|---|---|
| | | | | Forgetting Ratio ↓ | | |
| Fine-tuning | 57.51% | 85.04% | 51.33% | 7.15% | 4.81% | 0.00% |
| LwF (Li & Hoiem, 2017) | 54.79% | 72.52% | 59.32% | 2.76% | 6.92% | 0.00% |
| EWC (Kirkpatrick et al., 2017) | 62.47% | 46.55% | 61.55% | 9.95% | 13.42% | 0.00% |
| EWF (Xiao et al., 2023) | 69.65% | 92.51% | 79.07% | 0.47% | 1.03% | 0.00% |
| PathWeave (Yu et al., 2024) | 75.49% | 58.16% | 61.74% | 16.25% | 10.18% | 0.00% |
| BECAME (Li et al., 2025) | 72.82% | 92.70% | 66.04% | **-0.13%** | 3.36% | 0.00% |
| HiDe-LLaVA (Guo et al., 2025) | 68.89% | 91.93% | 70.51% | 0.00% | 1.03% | 0.00% |
| **MoInCL (Ours)** | **27.52%** | **9.18%** | **36.58%** | 0.07% | **-2.28%** | 0.00% |

The main results are shown in Tab. 1. We can see that our proposed MoInCL achieves state-of-the-art performance compared to all baseline methods. Except the average final accuracy of the Order 2, our method has the best performance on all three metrics across both orders. Specifically, in Order 1, our method surpasses the best baseline results by **16.25**, **1.71**, and **24.58** in terms of average final CIDEr score, average final accuracy, and average forgetting ratio, respectively. In Order 2, our method outperforms

Table 4: Forgetting ratio of each task in Order 2. Bold values denote the best results in each column, while underlined values indicate the second-best results in each column.

| Methods | Forgetting Ratio ↓ | | | | | |
| --- | --- | --- | --- | --- | --- | --- |
| | Flickr30k | MSR-VTT | MSVD-QA | OK-VQA | AudioCaps | Clotho-AQA |
| Fine-tuning | 93.02% | 85.72% | 31.23% | 49.77% | 68.06% | 0.00% |
| LwF (Li & Hoiem, 2017) | 91.20% | 85.83% | 31.51% | 40.07% | 60.60% | 0.00% |
| EWC (Kirkpatrick et al., 2017) | 91.08% | 92.21% | 40.49% | 37.91% | 70.31% | 0.00% |
| EWF (Xiao et al., 2023) | 89.86% | 78.28% | 6.04% | 4.15% | 54.89% | 0.00% |
| PathWeave (Yu et al., 2024) | 92.42% | 87.54% | 25.67% | 35.51% | 66.22% | 0.00% |
| BECAME (Li et al., 2025) | 90.50% | 80.31% | 15.48% | 9.92% | 74.29% | 0.00% |
| HiDe-LLaVA (Guo et al., 2025) | 89.85% | 81.52% | **1.74%** | 5.35% | 52.41% | 0.00% |
| **MoInCL (Ours)** | **22.04%** | **2.25%** | 2.60% | **3.33%** | **14.43%** | 0.00% |

the best baseline results by **37.21** and **37.71** for average final CIDEr score and average forgetting ratio, respectively.

The testing results of the first three incremental tasks (Image Captioning → Video Captioning → Video QA) are shown in Tab. 2. From these results, we observe that when the modality shift occurs from the Image Captioning task to the Video Captioning task, the performance of the previous task (Image Captioning) drops significantly across all baseline methods, with CIDEr score reductions ranging from 8.34 to 23.63. Additionally, when the task type shift occurs from the Video Captioning task to the Video QA task, the performance of the previous task (Video Captioning) also decreases significantly, with CIDEr score reductions ranging from 35.61 to 41.66. These results further validate our insight that both modality shift and task type shift directly contribute to the catastrophic forgetting problem, underscoring the core challenges of our proposed MICL scenario. For our method, the performance drop for the Image Captioning task is only **3.91** when the modality shift occurs. Moreover, we observe that the performance of the Video Captioning task improves after training on the Video QA task which introduces the task type shift issue. These findings further highlight the effectiveness of our method in mitigating the catastrophic forgetting problem in MICL by addressing both modality shift and task type shift challenges. For detailed results of each task and qualitative analysis, please refer to F, D, and G in Appendix.

We present the forgetting ratio of each task in both orders in Tab. 3 and 4, from which we can see that, our method outperforms baseline methods significantly, further demonstrating the superiority of our proposed method in mitigating the catastrophic forgetting in our proposed MICL scenario.

In Tab. 3, we observe that the Image Captioning task exhibits the largest reduction in forgetting ratio. This can be explained by the task order in Order 1: Image Captioning is learned relatively early and is subsequently followed by three consecutive QA tasks (including Image QA), resulting in a prolonged shift from captioning-style generation to QA-style prediction, in addition to modality shifts. And the final Image QA task reuses the image modality and partially reduces modality discrepancy, the sustained task-type shift remains the dominant source of interference for captioning. Among these factors, task-type shift is particularly disruptive to captioning, as later QA-oriented training shifts the generative behavior of the shared LLM backbone and, in the case of Image QA, also adapts the image-specific modality projector, both of which can interfere with previously learned captioning behaviors. Since PTGM explicitly preserves captioning-style behaviors through cross-task-type supervision and IKD stabilizes the shared LLM representation, Image Captioning — being highly sensitive to changes in the model's generative distribution — benefits most visibly from these mechanisms.

Moreover, in Order 1, we observe that under MoInCL, Image QA exhibits positive transfer after training the final task, Video Captioning. This occurs because Image QA is followed by Video Captioning. During Video Captioning training, PTGM generates pseudo QA pairs conditioned on caption targets, thereby reintroducing QA-style supervision even when the primary task objective is captioning. This mechanism helps maintain and reinforce QA behaviors in the shared LLM, which can result in mild positive transfer on the previously learned Image QA task. Furthermore, although modality shift (image → video) occurs, both tasks operate

within the visual domain. Stabilization of the shared LLM through IKD further reduces cross-modality interference, contributing to the observed behavior.

## 4.3 Ablation Studies

Table 5: Ablation results on the two task orders on each key component of our MoInCL. Bold values indicate the best results in each column, while underlined values represent the second-best results in each column.

| Methods | Order 1 | | | Order 2 | | |
| --- | --- | --- | --- | --- | --- | --- |
| | Avg. CIDEr ↑ | Avg. Acc. ↑ | Avg. Forget. ↓ | Avg. CIDEr ↑ | Avg. Acc. ↑ | Avg. Forget. ↓ |
| MoInCL w/o PTGM | 26.61 | 37.18 | 45.64% | 9.95 | **47.51** | 49.62% |
| MoInCL w/o IKD | 53.33 | 40.69 | 17.82% | 49.32 | 43.40 | 13.03% |
| MoInCL w/o WF | 39.12 | 39.51 | 37.70% | 33.55 | 35.90 | 43.23% |
| MoInCL | **55.31** | **42.29** | **14.21%** | **51.13** | 45.22 | **8.93%** |

To further assess the effectiveness of each key component in our proposed MoInCL, we conduct ablation studies on the Pseudo Target Generation Module (PTGM) and Instruction-based Knowledge Distillation (IKD) across two random task orders. The experimental results, presented in Tab. 5, clearly demonstrate that removing either PTGM or IKD leads to a performance drop in both task orders. This highlights the significance of each component in our framework. We also ablate the weight fusion (WF) mechanism. The results are shown in Tab. 5, which demonstrate that removing WF leads to consistent performance degradation across both task orders, indicating the effectiveness of integrating WF into our framework, with complementary benefits that are orthogonal to PTGM and IKD.

## 4.4 Analysis and Discussion

### 4.4.1 Results Analysis

We provide a more detailed analysis of the experimental results, specifically examining why the average accuracy of QA tasks in Order 2 does not achieve the best performance. In Order 2, the last four tasks follow the sequence: *Video QA → Image QA → Audio Captioning → Audio QA*, where QA tasks dominate. Consequently, the task type shift problem has a greater impact on captioning tasks than on QA tasks. For the baseline methods, as they focus less on addressing the task type shift problem, they prioritize QA tasks in the later stages of Order 2 rather than preserving knowledge from earlier tasks. This explains why most baseline methods perform better on QA tasks in Order 2 compared to Order 1. Nevertheless, our MoInCL still outperforms all other baselines in terms of average accuracy of QA tasks, except for EWF, where the difference is marginal. Additionally, MoInCL exhibits a lower average forgetting ratio compared to all baselines in both orders, and achieves lower forgetting ratio on each single task. Moreover, MoInCL maintains more stable performance across both task orders, further demonstrating its robustness.

Table 6: Experimental results on task transfer effectiveness. We evaluate modality transfer effectiveness within the same task type.

| Modality Transfer | | | | | |
| --- | --- | --- | --- | --- | --- |
| Video Cap | Image Cap → Video Cap | Audio QA | Video QA → Audio QA | Video QA | Image QA → Video QA |
| 47.12 | 48.03 | 58.28 | 59.94 | 44.81 | 45.74 |

Table 7: Experimental results on task transfer effectiveness. We evaluate modality transfer effectiveness within the same modality.

| Task Type Transfer | | | | | |
| --- | --- | --- | --- | --- | --- |
| Audio QA | Audio Cap → Audio QA | Image QA | Image Cap → Image QA | Video Cap | Video QA → Video Cap | Image Cap | Image QA → Image Cap |
| 58.28 | 61.75 | 35.00 | 36.50 | 47.12 | 51.25 | 77.5 | 81.93 |

### 4.4.2 Task Transfer Effectiveness

To investigate the mutual impact between different tasks, we evaluate the positive knowledge transfer across tasks that share the same modality or task type. Specifically, we conduct experiments to determine whether training on one task benefits a subsequent task within the same modality or task type. The experimental results are presented in Tab. 6 and 7. As shown, transferring the captioning ability from the image captioning task improves the CIDEr score of the video captioning task from 47.12 to 48.03. Similarly, transferring the question-answering capability from the video QA task enhances the accuracy of the audio QA task from 58.28 to 59.94. These results further demonstrate that transferring knowledge from a previous task to a new task with the same task type enhances the performance of this new task. Additionally, the audio QA ability is enhanced by transferring knowledge from the learned audio captioning task, improving accuracy from 58.28 to 61.75. Similarly, positive knowledge transfer is observed within the image and video modalities, further demonstrating the benefits of transferring knowledge across tasks within the same modality.

### 4.4.3 Analysis on the Computational Cost

For each experiment, *i.e.*, training a single baseline method or our MoInCL, we use a single RTX A6000 Ada GPU with 48GB of memory. Compared to the pure fine-tuning baseline, the average training time for our MoInCL increases by approximately 40% per epoch, while the inference time remains the same. For example, during training on the audio captioning task with the AudioCaps dataset, pure fine-tuning takes around 45 minutes per epoch, and our method requires approximately 64 minutes per epoch. Please note that, the reported ∼40% increase in training time accounts only for the additional optimization overhead introduced by IKD and PTGM during gradient updates. The generation of PTGM pseudo-labels (e.g., pseudo captions and pseudo QA pairs) is performed offline using frozen models prior to training each new task and is therefore not included in the per-epoch or per-iteration training time.

### 4.4.4 Analysis on the Corpus Size Sensitivity in IKD

We conduct an additional experiment on Order 1 to evaluate the sensitivity of IKD to the size of the Natural Instructions (NI) corpus. We evaluate different size of the NI corpus (0%, 50%, and 100%) for IKD. The results are presented in Tab. 8, which show that even with a substantially reduced instruction corpus (50%), MoInCL maintains comparable performance and forgetting reduction. While removing IKD degrades performance and increases forgetting, the performance gap between 50% and 100% NI remains relatively small under the current corpus setting. This suggests that, IKD does not exhibit strong sensitivity to corpus size. We note that broader domain coverage or larger instruction diversity may potentially provide additional benefits, but our results indicate that IKD does not critically depend on the full corpus to be effective.

Table 8: Experimental results on corpus size sensitivity in IKD. We evaluate different size of the Natural Instructions corpus used in IKD.

| Methods | Avg. CIDEr ↑ | Avg. Acc. ↑ | Avg. Forget. ↓ |
|---|---|---|---|
| w/o IKD (0% NI) | 53.33 | 40.69 | 17.82% |
| IKD with 50% NI | 55.04 | 42.17 | 14.96% |
| IKD with 100% NI | **55.31** | **42.29** | **14.21%** |

### 4.4.5 Evaluation of General LLM Capabilities

To evaluate whether continual learning under modality inconsistency affects the general reasoning capabilities of the LLM backbone, we conduct zero-shot evaluation on the MMLU (Hendrycks et al., 2021) benchmark before and after each continual learning step on Order 1 with our proposed MoInCL method. The evaluation results are shown in Tab 9. As shown, the LLM backbone's MMLU accuracy remains highly stable throughout the entire continual learning process, with fluctuations within ±1% of the initial performance, demonstrating no obvious catastrophic forgetting in the LLM's general reasoning capability. These results indicate that continual adaptation under MoInCL does not degrade the fundamental reasoning and knowledge capacity of

the LLM backbone. The LoRA-based adaptation together with IKD preserves general reasoning capability while enabling effective multimodal continual learning.

Table 9: Zero-shot evaluation of the LLM backbone before and after training on Order 1 with our MoInCL.

| Steps | 0 | 1 | 2 | 3 | 4 | 5 | 6 |
|---|---|---|---|---|---|---|---|
| Accuracy (%) | 43.87 | 43.27 | 43.01 | 43.41 | 43.42 | 43.13 | 43.34 |

## 5  Conclusion

In this paper, we explore the Modality-Inconsistent Continual Learning (MICL), a novel and practical continual learning scenario of Multimodal Large Language Models (MLLMs). To address the introduced MICL, we propose MoInCL, which incorporates a Pseudo Targets Generation Modul and an Instruction-based Knowledge Distillation constraint to mitigate the catastrophic forgetting caused by the inherent task type shift and modality shift problem in the context of MICL. Experiments on six multimodal incremental tasks demonstrate the effectiveness of our proposed MoInCL. This paper introduces a new direction for the continual learning of MLLMs.

**Broader Impact.** Our proposed continual modality-inconsistent continual learning allows the MLLMs to adapt to new modalities and task types without full retraining, which could enhance efficiency and privacy by reducing the need to transmit and store sensitive data.

## Limitations

Our Modality-Inconsistent Continual Learning (MICL) introduces a novel and practical continual learning scenario by incorporating inconsistent modalities and varying task types across incremental tasks. However, the scope of our work is constrained by the limited number of modalities (audio, image, and video) and task types (captioning and question-answering) included in the experiments. This restricts the generalizability of MICL to scenarios involving a broader range of modalities and task types. Another limitation lies in the pseudo QA pairs generated by PTGM, which may not fully capture the complete answer space of prior QA tasks, leading to incomplete supervision when mitigating the task type shift from QA to captioning tasks. These imperfect pseudo targets may thus still hinder a full resolution of the task type shift problem.

In the future, we plan to enhance our MICL framework by incorporating additional modalities, such as depth, 3D, or even joint inputs like joint audio-visual modalities. We also aim to introduce a broader range of task types, such as reasoning, grounding, decision-making, etc. Furthermore, scaling up MICL to larger datasets within each task is also a key objective to better enable the model to address the complexity and diversity of real-world multimodal tasks in continual learning.

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

## Appendix

In this appendix, we provide the process of generating three-round QA pairs from captions in Sec. A. We also include dataset details in Sec. B. Experimental results on additional task orders, upper bound results, larger models, detailed results of each task, qualitative analysis, and broader impacts are provided in Sec. C, D, E, F, G, and H, respectively.

## A  Three-Round QA Pairs Generation from Captions

Inspired by the question answering text generation process in (Panagopoulou et al., 2023), we adopt a similar three-round QA pair generation process from captions in our proposed Pseudo Targets Generation Module (PTGM). Given a caption from the dataset of the current captioning task $\mathcal{T}_i$, the objective is to generate a QA pair to address the task type shift problem when training on a captioning task within a seen modality. This process relies entirely on prompt engineering, where the caption is used as input to the pre-trained Large Language Model (LLM) component of our Multimodal Large Language Model (MLLM). Please note that, the LLM component employed in this process uses pre-trained weights, *i.e.*, the weights that are not fine-tuned on our incremental tasks.

In Round 1, the LLM takes an input with the format of:

```
Given the Mᵢ context:  "y", generate a potential short answer from it.
Provide just one or two words.  The answer words should be strictly
selected from the context.  Provide only the answer, nothing else.  Answer:,
```

where $M_i$ is the modality of the task $\mathcal{T}_i$, $\boldsymbol{y}$ denotes the sampled caption text. And the output of the LLM is used as the temporal short answer $\bar{\boldsymbol{y}}$.

In Round 2, the LLM takes the following prompt as input:

```
Given the Mᵢ context:  "y" and the answer:  "ȳ", generate a question
for the answer that can be inferred from the context.  Provide only one
question and nothing else.  Question:
```

and the output of the LLM in Round 2 is the question we aim to generate, which is denoted as $\tilde{\boldsymbol{t}}$.

Finally, in Round 3, the LLM processes the following prompt as input:

```
Answer the question using the given context.  The answer should be only one
or two words.  Context: "y".  Question: "t̃".  Answer:
```

and generates the final short answer $\tilde{y}$.

Based the above three rounds, the pseudo QA pair is obtained, where $\tilde{t}$ represents the pseudo question and $\tilde{y}$ denotes the pseudo answer.

For examples of generated QA pairs, please refer to Sec. G and Fig. 9.

## B  Dataset Details

Table 10: Details of the datasets used in our experiments.

| Task | Dataset | Sample number | | | |
|---|---|---|---|---|---|
| | | Total | Training | Validation | Testing |
| Image Captioning | Flickr30K | 31,784 | 29,783 | 1,000 | 1,000 |
| Image QA | OK-VQA | 14,055 | 8,007 | 1,002 | 5,046 |
| Audio Captioning | AudioCaps | 46,378 | 44,378 | 1,000 | 1,000 |
| Audio QA | Clotho-AQA | 10,480 | 6,181 | 1,823 | 2,476 |
| Video Captioning | MSR-VTT | 10,000 | 6,010 | 1,000 | 2,990 |
| Video QA | MSVD-QA | 50,476 | 30,904 | 6,415 | 13,157 |

In our experiments, we use the AudioCaps, Flickr30K, MSR-VTT, MSVD-QA, Clotho-AQA, and OK-VQA datasets for Audio Captioning, Image Captioning, Video Captioning, Video QA, Audio QA, and Image QA tasks, respectively. We summarize the details of these data in Tab. 10.

## C  Experimental Results on Additional Task Orders

Apart from the random task orders in Sec. 4.2, we also conduct additional experiments to further verify the effectiveness and robustness of our proposed MoInCL. Specifically, we construct a new random order: **Video Captioning → Image QA → Image Captioning → Video QA → Audio Captioning → Audio QA**, which we refer to as **Order 3**.

Additionally, we also manually create another task order: **Image QA → Video Captioning → Audio QA → Image Captioning → Video QA → Audio Captioning**, one of the most challenging task orders. This task order enforces frequent alternation between task types, following the pattern: $QA \rightarrow Captioning \rightarrow QA \rightarrow Captioning \rightarrow QA \rightarrow Captioning$, which ensures no two tasks of the same task type appear consecutively. Moreover, this order also introduces more frequent modality shifts, avoiding repetition of the same modality in adjacent tasks. This setting helps mitigate task-recency bias and offers a more rigorous evaluation of each method's ability to generalize under highly dynamic conditions. We refer to this extreme task order as **Order 4**.

The experimental results on these two new task orders are reported in Tab. 11. As shown, our method consistently achieves significant improvements over the baseline methods. Furthermore, its performance remains in line with the results on the original task orders, further highlighting the stability and robustness of our approach.

## D  Upper Bound Results

We present the testing results of the Upper Bound (joint training) on each task in Tab. 12.

Table 11: Experimental results on additional two task orders for different continual learning methods. Bold values indicate the best results in each column, while underlined values represent the second-best results in each column.

| Methods | Order 3 | | | Order 4 | | |
| --- | --- | --- | --- | --- | --- | --- |
| | Avg. CIDEr ↑ | Avg. Acc. ↑ | Avg. Forget. ↓ | Avg. CIDEr ↑ | Avg. Acc. ↑ | Avg. Forget. ↓ |
| Fine-tuning | 23.14 | 41.59 | 53.18% | 46.16 | 19.94 | 56.11% |
| EWF (Xiao et al., 2023) | 35.46 | 37.10 | 46.14% | 46.92 | 36.72 | 29.66% |
| PathWeave (Yu et al., 2024) | 28.46 | 40.50 | 51.73% | 47.27 | 20.54 | 53.72% |
| **MoInCL (Ours)** | **57.18** | **45.39** | **13.07%** | **57.77** | **40.81** | **14.93%** |
| Upper Bound (Joint training) | 66.69 | 48.97 | - | 66.69 | 48.97 | - |

Table 12: Experimental results of the Upper Bound (joint training) on each task.

| Methods | Flickr30k | MSR-VTT | MSVD-QA | OK-VQA | AudioCaps | Clotho-AQA |
| --- | --- | --- | --- | --- | --- | --- |
| Upper Bound (Joint training) | 80.24 | 54.76 | 48.54 | 38.16 | 65.07 | 60.22 |

# E  Experimental Results on Larger Models

To evaluate scalability on larger MLLMs, we conducted additional experiments using LLaMA-3.1-8B-Instruct as the LLM component (instead of LLaMA-3.2-1B-Instruct used in the main experiments). The experimental results are presented in Tab. 13, which clearly show that our method consistently outperforms the baselines, demonstrating the effectiveness and robustness of our method when scaling the LLM size from 1B to 8B parameters.

# F  Detailed Results of Each Task in Both Orders

We also present the detailed testing results for each task across the incremental steps in both orders in Tab. 14 and 15. These results show that our proposed MoInCL exhibits less performance drop compared to the baseline methods, demonstrating its superior ability to address catastrophic forgetting in the proposed Modality-Inconsistent Continual Learning (MICL) scenario.

# G  Qualitative Analysis

We present the qualitative results of the Fine-tuning, LwF (Li & Hoiem, 2017), EWC (Kirkpatrick et al., 2017), EWF (Xiao et al., 2023), PathWeave (Yu et al., 2024), and our MoInCL in Fig. 3, 4, 5, 6, 7, and 8, respectively. From these results, we can see that our MoInCL can generate better results with the incremental step increases, demonstrating the better capability in mitigating the catastrophic forgetting problem in our proposed Modality-Inconsistent Continual Learning (MICL) scenario. We also present the qualitative results of the generated pseudo QA pairs of the Image Captioning dataset (Flickr30k) used in our experiments. The

Table 13: Experimental results on Order 1 for different continual learning methods using LLaMA-3.1-8B-Instruct as the LLM component. Bold values indicate the best results in each column.

| Methods | Avg. CIDEr | Avg. Acc. | Avg. Forget. |
| --- | --- | --- | --- |
| Fine-tuning | 30.42 | 50.07 | 39.02% |
| LwF | 33.15 | 50.31 | 38.46% |
| EWC | 36.88 | 46.82 | 37.37% |
| EWF | 28.65 | 50.92 | 37.47% |
| PathWeave | 35.72 | 46.62 | 40.55% |
| **MoInCL (Ours)** | **65.24** | **52.94** | **9.21%** |

qualitative results are presented in Fig. 9. From these randomly sampled examples, we observe that the frozen LLM generally produces semantically consistent and visually grounded questions and answers. The generated questions align with the image captions and focus on relevant visual attributes such as object identity, actions, counting, and scene context.

## H   Broader Impacts

This work studies modality-inconsistent continual learning (MICL) for multimodal large language models (MLLMs), aiming to improve robustness under modality and task-type shifts. While such improvements can benefit adaptive multimodal systems (*e.g.*, accessibility tools, cross-modal assistants, and long-term interactive agents), broader societal and ethical considerations should be acknowledged.

**Bias and pseudo-target generation.**  In PTGM, pseudo-targets are generated offline before training each new task. Specifically, pseudo captions are produced by the previous-task model (used to initialize the current task), and pseudo QA pairs are generated by a frozen pretrained LLM. Both generators are pretrained on publicly available corpora, and our framework does not introduce additional data sources or new generative mechanisms beyond these foundation models. Therefore, MoInCL does not create new bias channels beyond those already present in widely used pretrained models. Nevertheless, as with any pseudo-labeling or LLM-based pipeline, imperfect or biased outputs may occur. Because pseudo-target generation is conducted offline, these outputs can be audited, filtered, or constrained (*e.g.*, through conservative decoding or confidence-based filtering) prior to training, which reduces the risk of amplifying undesirable patterns.

**Potential misuse.**  Like other multimodal modeling techniques, continual learning frameworks may be misused if deployed irresponsibly, for example in surveillance-oriented applications involving adaptive video or audio analysis. This work is intended for research purposes and benign applications such as accessibility support and multimodal reasoning systems. Deployment decisions remain the responsibility of practitioners and stakeholders.

**Energy and computational considerations.**  MoInCL introduces additional training overhead compared to standard fine-tuning due to the optimization of IKD and PTGM during gradient updates, as reported in Sec. 4.4.3. No additional model components are introduced during evaluation. We emphasize the importance of reporting computational cost and responsible use of resources when scaling multimodal systems.

**Dataset usage and ethical sourcing.**  All datasets used in this work are publicly available research benchmarks, and we follow their original licensing and usage terms. We do not collect new private data. Some datasets (*e.g.*, AudioCaps) may contain audio that includes personal or sensitive content; our work strictly uses the released benchmark data for research evaluation under the dataset's stated conditions.

Table 14: Detailed testing results for each task across the incremental steps in Order 1. The evaluation metric used for the AudioCaps, Flickr30K, and MSR-VTT datasets is CIDEr score, while that for the MSVD-QA, Clotho-AQA, and OK-VQA datasets is accuracy.

|  |  | AudioCaps | Flickr30K | MSVD-QA | Clotho-AQA | OK-VQA | MSR-VTT |
|---|---|---|---|---|---|---|---|
| Fine-tuning | Step 1 | 57.66 | - | - | - | - | - |
|  | Step 2 | 26.42 | 85.83 | - | - | - | - |
|  | Step 3 | 8.34 | 30.83 | 47.67 | - | - | - |
|  | Step 4 | 4.28 | 21.89 | 44.52 | 62.64 | - | - |
|  | Step 5 | 4.06 | 6.49 | 39.36 | 57.51 | 42.41 | - |
|  | Step 6 | 24.50 | 12.84 | 23.20 | 58.16 | 40.37 | 54.59 |
| LwF (Li & Hoiem, 2017) | Step 1 | 57.66 | - | - | - | - | - |
|  | Step 2 | 26.32 | 86.97 | - | - | - | - |
|  | Step 3 | 4.61 | 30.38 | 47.47 | - | - | - |
|  | Step 4 | 0.04 | 15.96 | 42.08 | 63.13 | - | - |
|  | Step 5 | 1.18 | 6.36 | 36.16 | 59.85 | 42.89 | - |
|  | Step 6 | 26.07 | 23.90 | 19.31 | 61.39 | 39.92 | 54.44 |
| EWC (Kirkpatrick et al., 2017) | Step 1 | 57.66 | - | - | - | - | - |
|  | Step 2 | 38.59 | 85.27 | - | - | - | - |
|  | Step 3 | 5.67 | 25.23 | 46.03 | - | - | - |
|  | Step 4 | 2.04 | 14.21 | 43.78 | 63.29 | - | - |
|  | Step 5 | 3.85 | 6.31 | 38.85 | 56.70 | 42.09 | - |
|  | Step 6 | 21.64 | 45.58 | 17.70 | 56.99 | 36.44 | 49.95 |
| EWF (Xiao et al., 2023) | Step 1 | 57.66 | - | - | - | - | - |
|  | Step 2 | 49.84 | 82.73 | - | - | - | - |
|  | Step 3 | 38.01 | 71.03 | 44.33 | - | - | - |
|  | Step 4 | 14.19 | 65.28 | 44.22 | 59.69 | - | - |
|  | Step 5 | 15.48 | 6.08 | 43.98 | 59.53 | 40.75 | - |
|  | Step 6 | 17.50 | 6.20 | 9.28 | 59.41 | 40.33 | 50.07 |
| PathWeave (Yu et al., 2024) | Step 1 | 59.86 | - | - | - | - | - |
|  | Step 2 | 13.54 | 82.32 | - | - | - | - |
|  | Step 3 | 2.95 | 12.02 | 46.00 | - | - | - |
|  | Step 4 | 0.54 | 9.07 | 37.28 | 63.13 | - | - |
|  | Step 5 | 4.19 | 6.26 | 28.97 | 57.84 | 42.42 | - |
|  | Step 6 | 14.67 | 34.44 | 17.60 | 52.87 | 38.10 | 53.48 |
| BECAME (Li et al., 2025) | Step 1 | 57.66 | - | - | - | - | - |
|  | Step 2 | 55.71 | 81.46 | - | - | - | - |
|  | Step 3 | 18.34 | 63.77 | 45.61 | - | - | - |
|  | Step 4 | 5.81 | 54.34 | 45.31 | 60.18 | - | - |
|  | Step 5 | 9.43 | 6.04 | 40.39 | 59.13 | 41.13 | - |
|  | Step 6 | 15.67 | 5.95 | 15.49 | 60.26 | 39.75 | 51.47 |
| HiDe-LLaVA (Guo et al., 2025) | Step 1 | 57.66 | - | - | - | - | - |
|  | Step 2 | 54.67 | 81.87 | - | - | - | - |
|  | Step 3 | 53.32 | 80.43 | 44.01 | - | - | - |
|  | Step 4 | 38.22 | 81.62 | 44.17 | 58.28 | - | - |
|  | Step 5 | 36.25 | 73.91 | 44.47 | 58.84 | 37.57 | - |
|  | Step 6 | 36.72 | 72.05 | 25.38 | 58.97 | 37.71 | 51.21 |
| MoInCL (Ours) | Step 1 | 57.66 | - | - | - | - | - |
|  | Step 2 | 56.58 | 81.15 | - | - | - | - |
|  | Step 3 | 56.51 | 82.71 | 43.38 | - | - | - |
|  | Step 4 | 43.44 | 81.91 | 43.43 | 57.71 | - | - |
|  | Step 5 | 43.01 | 74.19 | 43.51 | 57.51 | 40.75 | - |
|  | Step 6 | 41.79 | 73.70 | 27.51 | 57.67 | 41.68 | 50.44 |
| Upper Bound (Joint training) |  | 65.07 | 80.24 | 48.54 | 60.22 | 38.16 | 54.76 |

Table 15: Detailed testing results for each task across the incremental steps in Order 2. The evaluation metric used for the AudioCaps, Flickr30K, and MSR-VTT datasets is CIDEr score, while that for the MSVD-QA, Clotho-AQA, and OK-VQA datasets is accuracy.

|  | | Flickr30K | MSR-VTT | MSVD-QA | OK-VQA | AudioCaps | Clotho-AQA |
|---|---|---|---|---|---|---|---|
| Fine-tuning | Step 1 | 77.50 | - | - | - | - | - |
| | Step 2 | 64.04 | 48.03 | - | - | - | - |
| | Step 3 | 12.12 | 8.64 | 46.20 | - | - | - |
| | Step 4 | 5.86 | 8.23 | 39.38 | 37.13 | - | - |
| | Step 5 | 9.63 | 14.05 | 24.91 | 17.24 | 63.19 | - |
| | Step 6 | 5.41 | 6.86 | 31.77 | 18.65 | 20.18 | 60.62 |
| LwF (Li & Hoiem, 2017) | Step 1 | 77.50 | - | - | - | - | - |
| | Step 2 | 53.87 | 48.70 | - | - | - | - |
| | Step 3 | 10.20 | 7.80 | 47.64 | - | - | - |
| | Step 4 | 7.41 | 8.44 | 37.14 | 36.51 | - | - |
| | Step 5 | 12.51 | 18.08 | 31.44 | 19.47 | 59.37 | - |
| | Step 6 | 6.82 | 6.90 | 32.63 | 21.88 | 23.39 | 61.87 |
| EWC (Kirkpatrick et al., 2017) | Step 1 | 77.50 | - | - | - | - | - |
| | Step 2 | 62.65 | 47.73 | - | - | - | - |
| | Step 3 | 10.45 | 9.66 | 45.79 | - | - | - |
| | Step 4 | 7.19 | 7.85 | 37.42 | 35.90 | - | - |
| | Step 5 | 12.10 | 4.24 | 27.59 | 21.09 | 64.40 | - |
| | Step 6 | 6.91 | 3.72 | 27.25 | 22.29 | 19.12 | 63.41 |
| EWF (Xiao et al., 2023) | Step 1 | 77.50 | - | - | - | - | - |
| | Step 2 | 69.16 | 45.30 | - | - | - | - |
| | Step 3 | 56.10 | 9.69 | 45.33 | - | - | - |
| | Step 4 | 8.26 | 9.85 | 44.74 | 34.95 | - | - |
| | Step 5 | 8.04 | 10.24 | 43.31 | 33.10 | 53.36 | - |
| | Step 6 | 7.86 | 9.84 | 42.59 | 33.50 | 24.07 | 61.47 |
| PathWeave (Yu et al., 2024) | Step 1 | 77.22 | - | - | - | - | - |
| | Step 2 | 53.60 | 50.01 | - | - | - | - |
| | Step 3 | 7.36 | 8.35 | 47.87 | - | - | - |
| | Step 4 | 6.99 | 7.14 | 41.17 | 36.38 | - | - |
| | Step 5 | 8.01 | 7.86 | 33.89 | 22.27 | 62.90 | - |
| | Step 6 | 5.85 | 6.23 | 35.58 | 23.46 | 21.25 | 64.34 |
| BECAME (Li et al., 2025) | Step 1 | 77.50 | - | - | - | - | - |
| | Step 2 | 77.22 | 47.64 | - | - | - | - |
| | Step 3 | 52.16 | 9.82 | 47.35 | - | - | - |
| | Step 4 | 7.24 | 9.59 | 46.36 | 34.48 | - | - |
| | Step 5 | 8.04 | 8.81 | 43.11 | 31.62 | 58.74 | - |
| | Step 6 | 7.36 | 9.38 | 40.02 | 31.06 | 15.10 | 58.52 |
| HiDe-LLaVA (Guo et al., 2025) | Step 1 | 77.50 | - | - | - | - | - |
| | Step 2 | 77.19 | 48.97 | - | - | - | - |
| | Step 3 | 78.20 | 9.37 | 45.19 | - | - | - |
| | Step 4 | 7.92 | 9.65 | 45.11 | 34.71 | - | - |
| | Step 5 | 8.03 | 8.89 | 44.88 | 33.11 | 52.29 | - |
| | Step 6 | 7.87 | 9.05 | 44.40 | 32.85 | 24.88 | 61.19 |
| MoInCL (Ours) | Step 1 | 77.50 | - | - | - | - | - |
| | Step 2 | 73.59 | 48.03 | - | - | - | - |
| | Step 3 | 70.88 | 48.34 | 43.11 | - | - | - |
| | Step 4 | 63.32 | 47.56 | 42.27 | 33.35 | - | - |
| | Step 5 | 61.91 | 47.78 | 42.24 | 33.46 | 53.79 | - |
| | Step 6 | 60.42 | 46.95 | 41.99 | 32.24 | 46.03 | 61.43 |
| Upper Bound (Joint training) | | 80.24 | 54.76 | 48.54 | 38.16 | 65.07 | 60.22 |

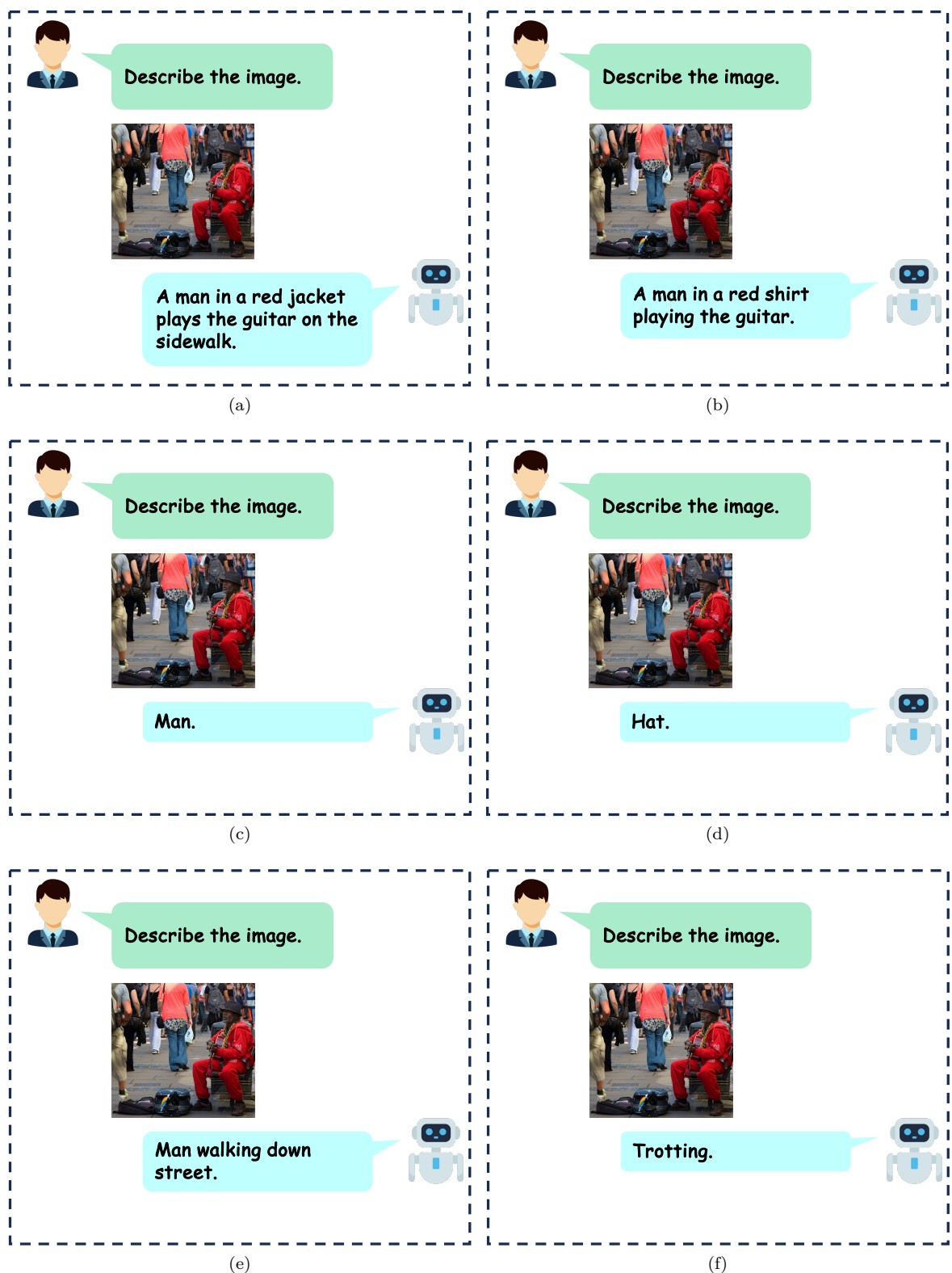

Figure 3: Qualitative results of the Fine-tuning method in Order 2. The sample is randomly selected from the test set of Task 1 (Image Captioning). The results are generated using models trained after after (a) Task 1, (b) Task 2, (c) Task 3, (d) Task 4, (e) Task 5, and (f) Task 6.

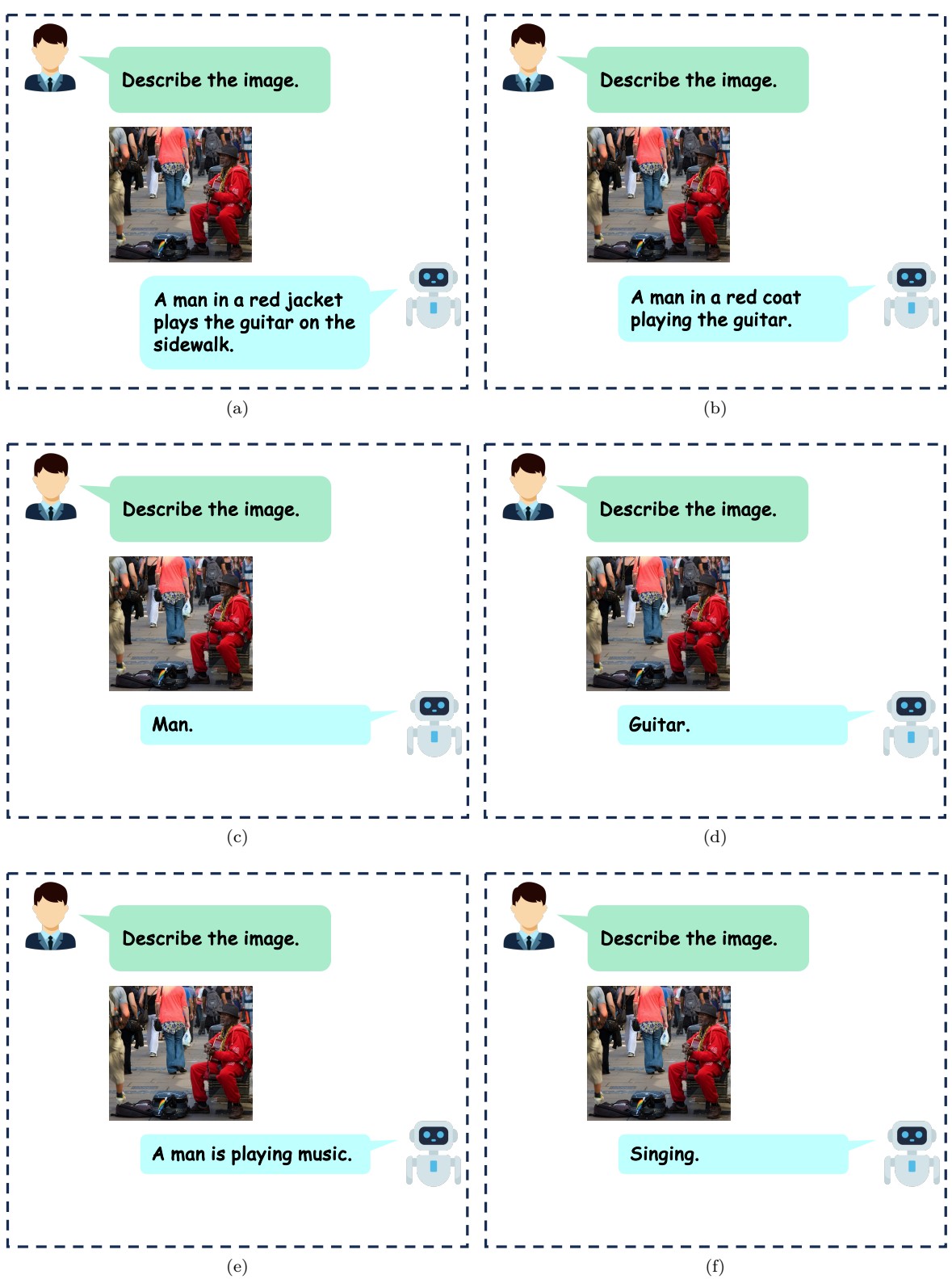

Figure 4: Qualitative results of the LwF (Li & Hoiem, 2017) method in Order 2. The sample is randomly selected from the test set of Task 1 (Image Captioning). The results are generated using models trained after after (a) Task 1, (b) Task 2, (c) Task 3, (d) Task 4, (e) Task 5, and (f) Task 6.

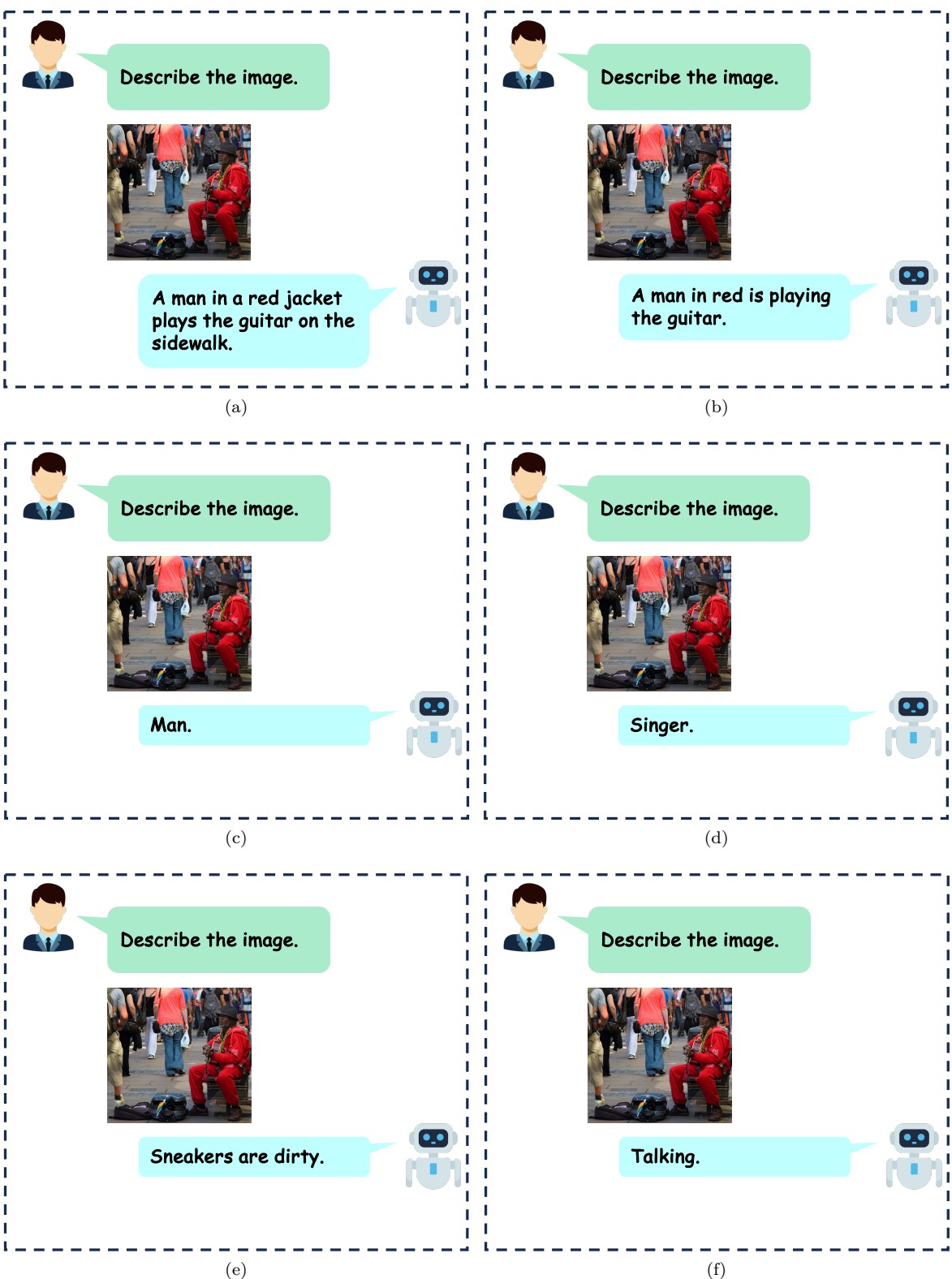

Figure 5: Qualitative results of the EWC (Kirkpatrick et al., 2017) method in Order 2. The sample is randomly selected from the test set of Task 1 (Image Captioning). The results are generated using models trained after after (a) Task 1, (b) Task 2, (c) Task 3, (d) Task 4, (e) Task 5, and (f) Task 6.

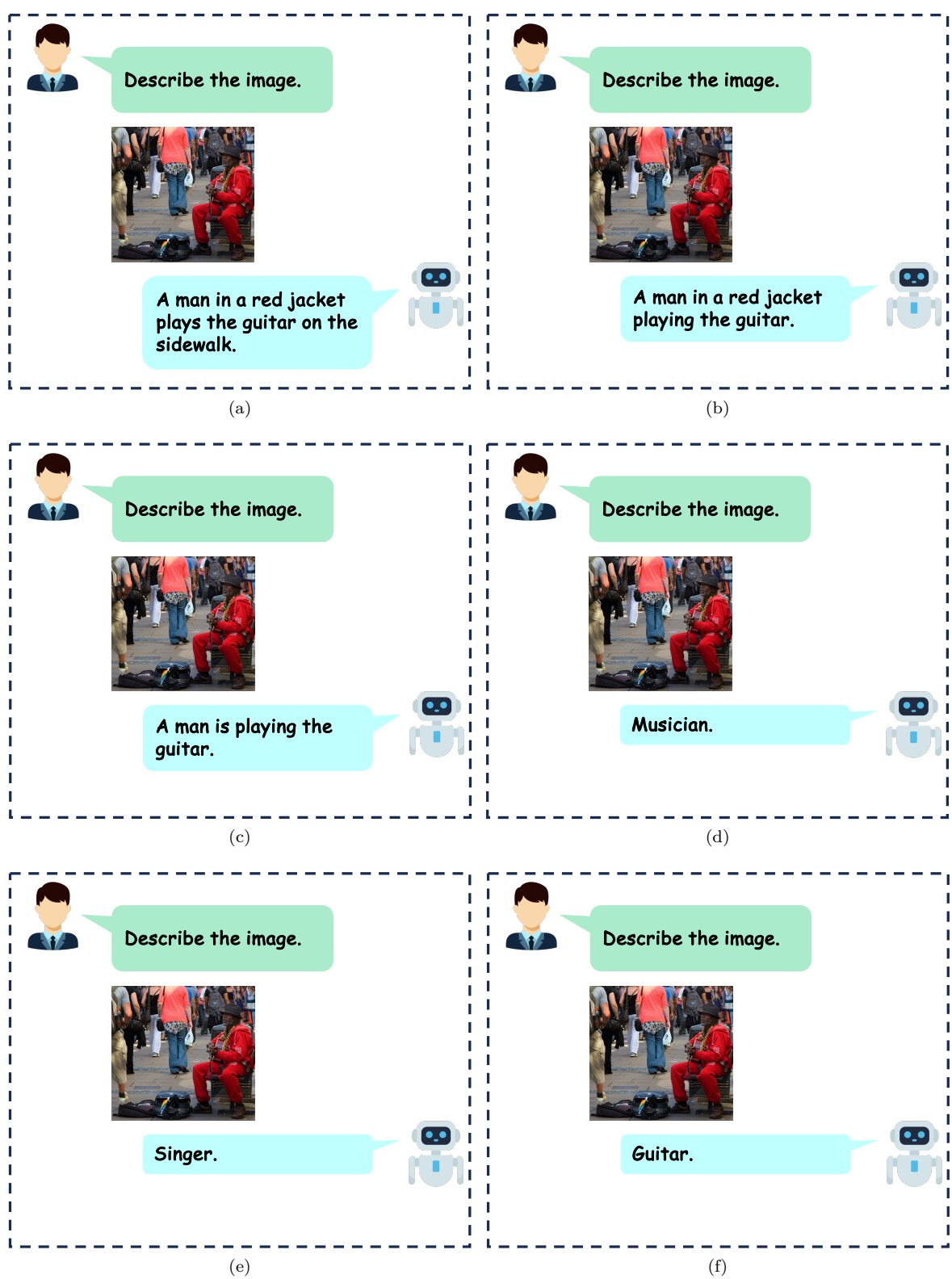

Figure 6: Qualitative results of the EWF (Xiao et al., 2023) method in Order 2. The sample is randomly selected from the test set of Task 1 (Image Captioning). The results are generated using models trained after after (a) Task 1, (b) Task 2, (c) Task 3, (d) Task 4, (e) Task 5, and (f) Task 6.

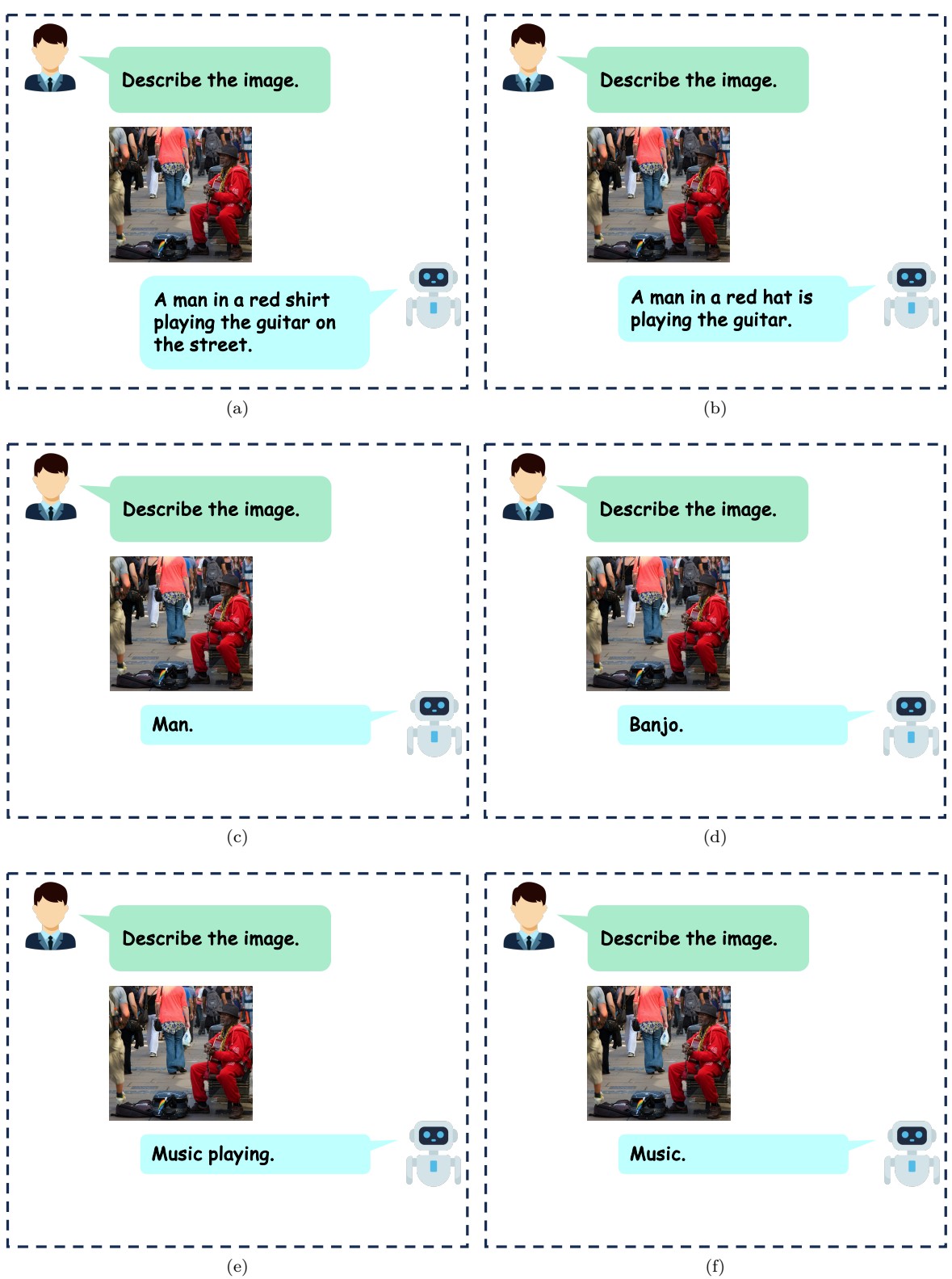

Figure 7: Qualitative results of the PathWeave (Yu et al., 2024) method in Order 2. The sample is randomly selected from the test set of Task 1 (Image Captioning). The results are generated using models trained after after (a) Task 1, (b) Task 2, (c) Task 3, (d) Task 4, (e) Task 5, and (f) Task 6.

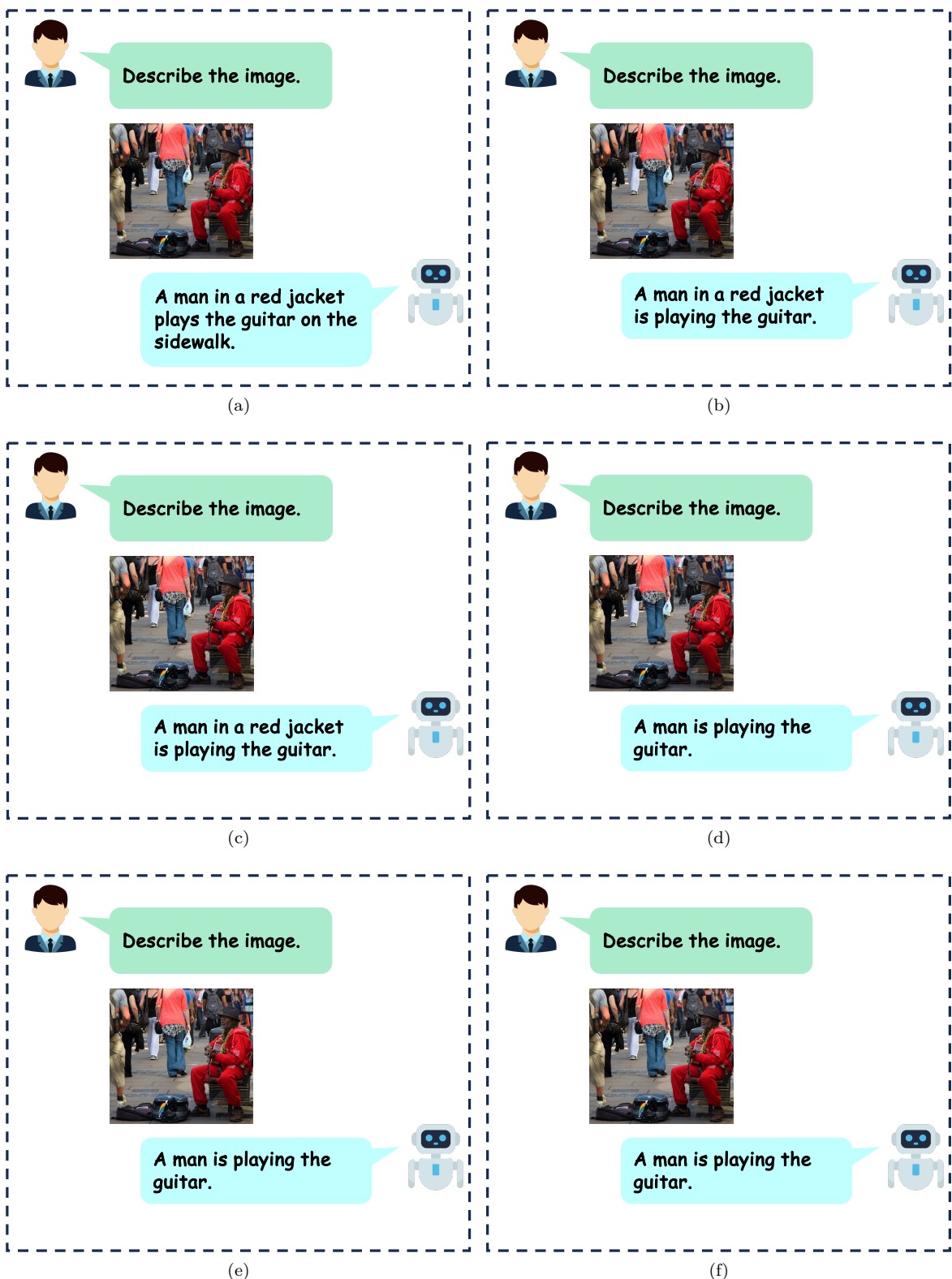

Figure 8: Qualitative results of our proposed MoInCL in Order 2. The sample is randomly selected from the test set of Task 1 (Image Captioning). The results are generated using models trained after after (a) Task 1, (b) Task 2, (c) Task 3, (d) Task 4, (e) Task 5, and (f) Task 6.

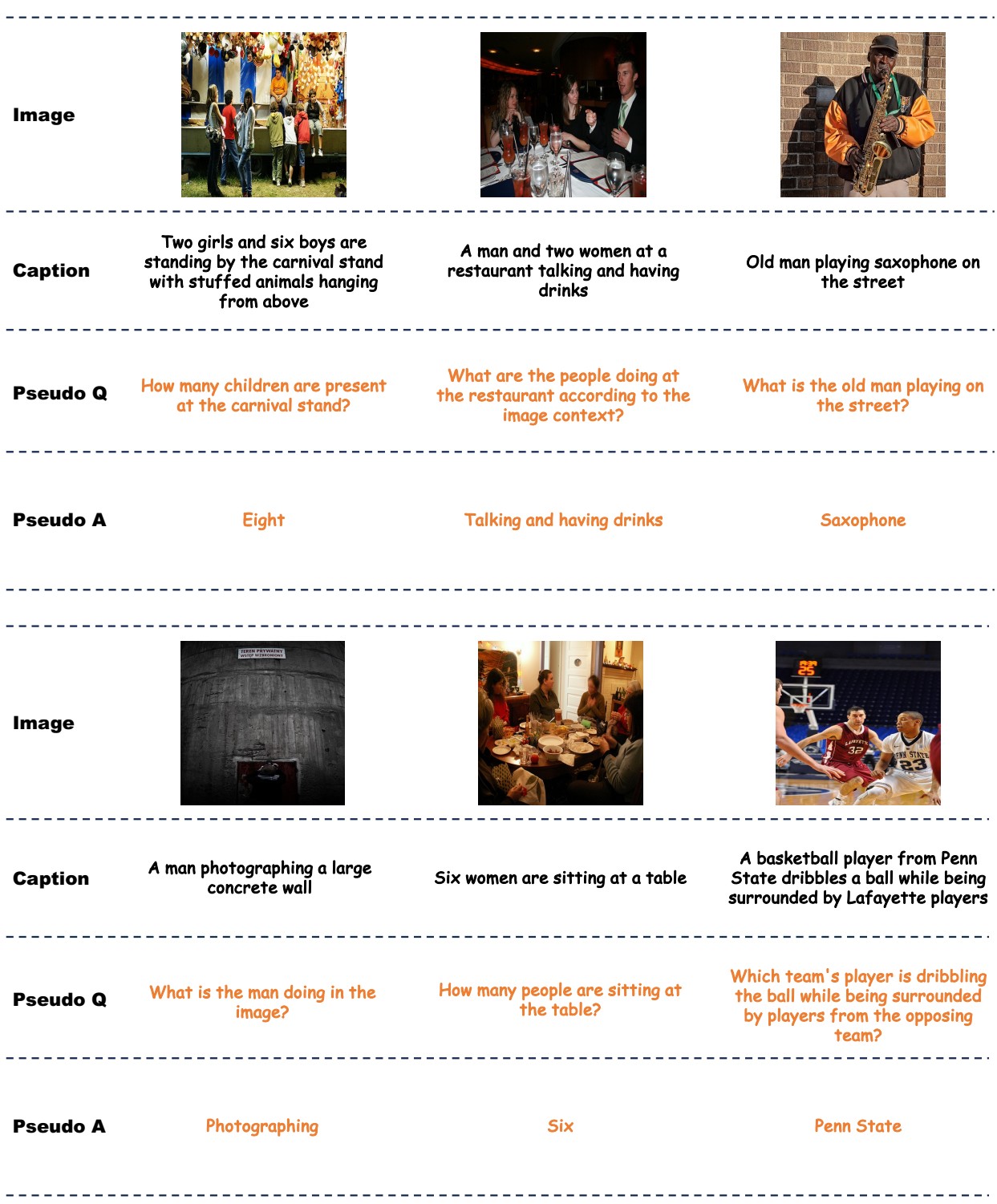

Figure 9: Qualitative results of generated pseudo QA pairs of the Image Captioning dataset.

