# OpenReview forum: "Modality-Inconsistent Continual Learning of Multimodal Large Language Models"
_TMLR — Accepted by TMLR_

### Review · Reviewer_DYDZ · 2026-02-08

**Summary Of Contributions:**

This paper introduces a new continual learning scenario called Modality-Inconsistent Continual Learning (MICL) for Multimodal Large Language Models (MLLMs), where tasks incrementally introduce inconsistent modalities (image, audio, video) combined with varying task types (captioning or question-answering). This setup aims to simulate real-world challenges where both modality shifts and task type shifts contribute to catastrophic forgetting, going beyond prior vision-only or modality-incremental settings.

**Additional Comments:**

Overall, this is a promising direction with a well-motivated scenario, but the truncation limits full assessment—assuming the full paper includes results, it could be strong with revisions. Consider open-sourcing the benchmark code for reproducibility. The weight fusion (Eq. 7) is downplayed as non-novel, but its integration is effective; perhaps ablate it separately.

I believe the introduced method is novel and would be of interest to relevant people.

**Audience:**

Yes

**Audience Explanation:**

The findings on handling modality-inconsistent shifts could inform practical applications like adaptive assistants processing diverse inputs (e.g., voice, video). Even if the method has room for improvement, the benchmark on six multimodal tasks provides a useful starting point for the community, especially as MLLMs like LLaVA and Qwen-VL gain traction.

**Broader Impact Concerns:**

The work focuses on improving continual learning in MLLMs, which could enable more efficient, adaptive AI systems for applications like assistive tech (e.g., describing audio/video for accessibility). However, if not addressed, biases in pseudo target generation (e.g., from LLM hallucinations) could propagate stereotypes across modalities, especially in captioning/QA for diverse content. The paper lacks a Broader Impact Statement—add one discussing potential misuse (e.g., in surveillance via continual video/audio adaptation) and energy costs of training multimodal models. Also, note ethical sourcing of datasets (e.g., AudioCaps may include sensitive audio).

**Claims And Evidence:**

No

**Claims Explanation:**

While the conceptual claims about MICL's novelty and the challenges of combined shifts are reasonably supported by the problem formulation and related work discussion, the empirical claims of MoInCL's superiority are not fully convincing due to the truncated document. Only setup details are provided (e.g., datasets like COCO for image captioning, AudioCaps for audio), but no actual results, tables, or ablation studies are shown in the visible pages. This makes it impossible to verify metrics like average performance or forgetting rates across the six tasks. Additionally, evidence for why PTGM and IKD outperform baselines (e.g., via qualitative examples or loss analyses) is described abstractly but lacks concrete data. The distinction from methods like LwF is argued, but without side-by-side empirical comparisons in the text, it's not clear whether the improvements are statistically significant or robust to hyperparameter variation. More rigorous evidence, such as error bars, multiple seeds, or sensitivity analyses, would be needed to strengthen the claims.

**Requested Changes:**

1. Include all tables, figures, and metrics from the experiments (e.g., performance on each task after incremental training, average accuracy, backward transfer). Since the document is truncated, ensure the full paper has these, with ablations on PTGM (e.g., without three-round generation) and IKD (e.g., varying instruction set size). Run experiments with multiple random seeds and report variances to confirm robustness.

2. Elaborate on the PTGM's QA pair generation prompts (currently in appendix—include or summarize inline). Provide more examples of pseudo inputs/targets. In Eq. (3), specify how λ_i and λ'_i are chosen (e.g., hyperparameter search?). Distinguish IKD more clearly from standard KD with quantitative evidence of why pure text instructions avoid constraining new task learning.

3. Compare against more recent multimodal CL works like HiDe-LLaVA (Guo et al., 2025) or CL-MoE (Huai et al., 2025), not just classics like EWC/LwF. Include modality-incremental baselines from Yu et al. (2024) with task type variations. Discuss failure cases or limitations, e.g., scalability to more modalities/tasks.

4. Fix minor typos (e.g., "MoInCL" vs. "MolnCL" in some places) and inconsistent notation (e.g., F_Θ vs. \mathcal{F}_\Theta). In Fig. 2, label components more precisely. Expand Sec. 3.1 to discuss why only captioning/QA are chosen and how other tasks (e.g., localization) fit.

5.  Test on larger MLLMs (e.g., beyond LoRA-tuned base) or real-world sequences (e.g., non-fixed task order). Analyze computational overhead of PTGM/IKD.

---

> ### Author Response · Authors · 2026-02-27
> **Response to Reviewer DYDZ**
>
> We sincerely thank the Reviewer DYDZ for the constructive feedback! We have addressed your concerns below and revised our paper accordingly, and the updated parts are highlighted using blue color. If there are any additional questions, we are willing to address them and revise our paper.
>
> > ### **Q1: Inaccurate, Unconvincing and Unclear Evidence Due to Document Truncation. Include all tables, figures, and metrics from the experiments. Ablation studies. Robustness.**
>
> We would like to clarify that the submitted manuscript is the **full document**, including all sections, figures, tables, and appendices, **without any truncation**.
> All experimental comparisons over baselines, including average final performance and forgetting rates across six tasks, are reported in Table 1 (Sec. 4.2). Per-task forgetting ratios are presented in Tables 3 and 4, and ablation studies validating the contributions of PTGM and IKD are provided in Sec. 4.3.
> In addition, robustness is evaluated under multiple task orders (two in the main paper and additional orders in the appendix), demonstrating consistent superiority of MoInCL under varying modality and task-type shifts.
> These results provide direct, side-by-side empirical comparisons with representative and state-of-the-art baselines, clearly supporting the empirical claims of MoInCL’s effectiveness.
>
> > ### **Q2: Elaborate on the PTGM's QA pair generation prompts. Provide more examples of pseudo inputs/targets. How $\lambda_i$ and $\lambda'_i$ are chosen. Distinguish IKD from standard KD on why pure text instructions avoid constraining new task learning.**
>
> Thank you for the suggestion. The detailed prompt templates for the three-round pseudo QA generation in PTGM are provided in Appendix A. We have also included examples of generated pseudo QA pairs in Appendix G (Fig. 9).
> Regarding the hyperparameters $\lambda_i$ and $\lambda'_i$ in Eq. (3), we set them to 1.0 and 0.5, respectively.
>
> Concerning the distinction between IKD and standard KD, traditional KD methods (e.g., LwF) align logits using the same multimodal training samples, which can overly constrain adaptation to new modalities in the MICL scenario.
>
> In contrast, our IKD operates purely at the language level by aligning the outputs of the LLM component on a fixed pure text instruction set. Since modality encoders are frozen and the source of forgetting under modality shift lies in updates to the LLM component, IKD stabilizes the LLM’s global generative prior without imposing constraints on modality-specific feature adaptation, thereby preserving previously learned capabilities for old modalities and mitigating forgetting caused by modality shift while still allowing effective learning of new modalities. Empirically, as shown in Sec. 4.3 (Table 5), removing IKD leads to increased forgetting and degraded performance, validating its effectiveness.
>
> > ### **Q3: Compare against more recent multimodal CL works. Include modality-incremental baselines from Yu et al. (2024) with task type variations. Discuss failure cases or limitations.**
>
> Thank you for the suggestion. We would like to clarify that our experiments already include recent baselines. Specifically, we compare against BECAME [1] and the modality-incremental method of Yu et al. [2] with task-type variations in Sec. 4.2 (Table 1).
>
> In addition, following the reviewer's suggestion, we further include experimental comparisons with HiDe-LLaVA [3], one of the most recent continual learning methods for MLLMs. The results are summarized below.
>
> Order 1:
> | Methods | Avg. CIDEr $\uparrow$  | Avg. Acc. $\uparrow$ | Avg. Forget. $\downarrow$ |
> |-|-|-|-|
> | HiDe-LLaVA [3] | 25.27 | 38.35 | 46.47%|
> |**MoInCL**|**55.31**|**42.29**|**14.21%**|
>
> Order 2:
> | Methods | Avg. CIDEr $\uparrow$| Avg. Acc. $\uparrow$ | Avg. Forget. $\downarrow$ |
> |-|-|-|-|
> | HiDe-LLaVA [3] | 13.93 | **46.15** | 46.18% |
> | **MoInCL** | **51.13** | 45.22 | **8.93%** |
>
> MoInCL consistently achieves significantly lower forgetting and substantially stronger generation performance across task orders, demonstrating its effectiveness compared with recent multimodal CL approaches. We have incorporated this new comparison in our revised manuscript (Tables 1, 3, 4, 14, and 15).
>
> Regarding limitations, a dedicated Limitations section is provided at the end of the main paper. We explicitly discuss (i) the restricted number of modalities (audio/image/video) and task types (captioning/QA) evaluated in this work, (ii) the potential incompleteness of pseudo QA supervision generated by PTGM, and (iii) scalability to broader modality and task settings. These aspects outline the current boundaries of MICL and possible future extensions.
>
> [1] BECAME: Bayesian Continual Learning with Adaptive Model Merging. In *ICML* 2025.
>
> [2] LLMs Can Evolve Continually on Modality for X-Modal Reasoning. In *NeurIPS* 2024.
>
> [3] HiDe-LLaVA: Hierarchical Decoupling for Continual Instruction Tuning of Multimodal Large Language Model. In *ACL* 2025.

---

> > ### Author Response · Authors · 2026-02-27
> > **Response to Reviewer DYDZ (continued)**
> >
> > > ### **Q4: Expand Sec. 3.1 to discuss why only captioning/QA are chosen and how other tasks fit.**
> >
> > Thank you for your suggestion. In this work, we focus on two task types (captioning and question-answering) because they are among the most commonly studied in multimodal and continual learning scenarios [1, 2], and they cover both generative and discriminative language outputs. Following common practice, we adopt these tasks to establish a standardized and comparable benchmark setting.
> >
> > Additionally, most of multimodal task types such as audio/video event localization, temporal language grounding, vision-language navigation can often be reformulated into question-answering formats (e.g., "When does the event occur?" or "Which region corresponds to the query?"). Therefore, the choice of captioning and QA serves as a representative and flexible interface for evaluating modality-inconsistent continual learning.
> >
> > We have highlighted and clarified this discussion in Sec. 3.1 in the revised manuscript.
> >
> > [1] Continual Instruction Tuning for Large Multimodal Models. arXiv:2311.16206.
> >
> > [2] LLMs Can Evolve Continually on Modality for X-Modal Reasoning. In *NeurIPS* 2024.
> >
> > > ### **Q5: Test on larger MLLMs. Real-world sequences (e.g., non-fixed task order). Analyze computational overhead.**
> >
> > Thank you for your suggestion. To evaluate scalability on larger MLLMs, we conducted additional experiments using LLaMA-3.1-8B-Instruct as the LLM component (instead of LLaMA-3.2-1B-Instruct used in the main experiments). The experimental results are presented in the following table and clearly show that our method consistently outperforms the baselines, demonstrating the effectiveness and robustness of our method when scaling the LLM size from 1B to 8B parameters.
> >
> > | Methods| Avg. CIDEr $\uparrow$ | Avg. Acc. $\uparrow$| Avg. Forget. $\downarrow$|
> > |-|-|-|-|
> > |Fine-tuning|30.42|50.07|39.02%|
> > |LwF|33.15|50.31|38.46%|
> > |EWC|36.88|46.82|37.37%|
> > |EWF|28.65|50.92|37.47%|
> > |PathWeave|35.72|46.62|40.55%|
> > |**MoInCL**|**65.24**|**52.94**|**9.21%**|
> >
> > We have included these results in Appendix E in our revised manuscript.
> >
> > Regarding task order sequences, our experiments are not restricted to a single fixed order. We evaluate multiple task orders (two in the main paper and additional two in the appendix), simulating varying modality and task-type sequences to better reflect real-world continual learning scenarios.
> >
> > Regarding computational overhead, we provide a dedicated analysis in Sec. 4.4.3. Compared to pure fine-tuning, MoInCL increases training time by approximately 40% per epoch due to the additional optimization of IKD and PTGM during gradient updates. Importantly, pseudo-label generation in PTGM is performed offline using frozen models prior to training each new task and is therefore not included in the per-epoch training time. The inference time remains unchanged, as our method does not introduce additional model components during evaluation.

---

> > > ### Author Response · Authors · 2026-02-27
> > > **Response to Reviewer DYDZ (continued)**
> > >
> > > > ### **Q6: Broader Impact Concerns.**
> > >
> > > Thank you for raising these broader impact considerations.
> > >
> > > **Bias and pseudo-target generation.**
> > > In PTGM, pseudo-targets are generated offline before training each new task. Specifically, pseudo captions are produced by the previous-task model (used to initialize the current task), and pseudo QA pairs are generated by a frozen pretrained LLM. Both generators are pretrained on publicly available corpora, and our framework does not introduce additional data sources or new generative mechanisms beyond these foundation models. Therefore, MoInCL does not create new bias channels beyond those already present in widely used pretrained models.
> > >
> > > Nevertheless, as with any pseudo-labeling or LLM-based pipeline, imperfect or biased outputs may occur. Because pseudo-target generation is conducted offline, these outputs can be audited, filtered, or constrained (e.g., through conservative decoding or confidence-based filtering) prior to training, which reduces the risk of amplifying undesirable patterns.
> > >
> > > **Potential misuse.**
> > > Like other multimodal modeling techniques, continual learning frameworks may be misused if deployed irresponsibly, for example in surveillance-oriented applications involving adaptive video or audio analysis. This work is intended for research purposes and benign applications such as accessibility support and multimodal reasoning systems. Deployment decisions remain the responsibility of practitioners and stakeholders.
> > >
> > > **Energy and computational considerations.**
> > > MoInCL introduces additional training overhead compared to standard fine-tuning due to the optimization of IKD and PTGM during gradient updates, as reported in Sec. 4.4.3. No additional model components are introduced during evaluation. We emphasize the importance of reporting computational cost and responsible use of resources when scaling multimodal systems.
> > >
> > > **Dataset usage and ethical sourcing.**
> > > All datasets used in this work are publicly available research benchmarks, and we follow their original licensing and usage terms. We do not collect new private data. Some datasets (e.g., AudioCaps) may contain audio that includes personal or sensitive content; our work strictly uses the released benchmark data for research evaluation under the dataset's stated conditions.
> > >
> > > This Broader Impact discussion has been incorporated into Appendix H in the revised manuscript.
> > >
> > > > ### **Q7: Open-sourcing the benchmark code.**
> > >
> > > Thank you for the suggestion. We will release our code to ensure full reproducibility upon acceptance.
> > >
> > > > ### **Q8: Ablation study on weight fusion.**
> > >
> > > Thank you for your suggestion. We have conducted the ablation study on the weight fusion (WF). The results are presented in the following tables:
> > >
> > > Order 1:
> > > | Methods                        | Avg. CIDEr           $\uparrow$  | Avg. Acc.           $\uparrow$ | Avg. Forget.          $\downarrow$ |
> > > |--------------------------------|----------------------------------|--------------------------------|------------------------------------|
> > > | MoInCL w/o WF                  | 39.12                            | 39.51                          | 37.70%                             |
> > > | MoInCL                         | 55.31                            | 42.29                          | 14.21%                             |
> > >
> > >
> > > Order 2:
> > > | Methods                        | Avg. CIDEr           $\uparrow$  | Avg. Acc.           $\uparrow$ | Avg. Forget.          $\downarrow$ |
> > > |--------------------------------|----------------------------------|--------------------------------|------------------------------------|
> > > | MoInCL w/o WF                  | 33.55                            | 35.90                          | 43.23%                             |
> > > | MoInCL                         | 51.13                            | 45.22                          | 8.93%                              |
> > >
> > > This ablation study demonstrates that removing WF leads to consistent performance degradation across both task orders, indicating the effectiveness of integrating WF into our framework, with complementary benefits that are orthogonal to PTGM and IKD.
> > >
> > > We have included this ablation study in Sec. 4.3 of our revised manuscript.

---

### Review · Reviewer_VK5i · 2026-02-11

**Summary Of Contributions:**

Here is a revised draft of the TMLR review. It incorporates your specific feedback regarding the missing LoRA-based baselines while maintaining the positive assessment of the benchmark and problem formulation.

Summary of Contributions
The paper introduces "Modality-Inconsistent Continual Learning" (MICL), a novel and practical continual learning scenario for Multimodal Large Language Models (MLLMs). Unlike standard modality-incremental settings, MICL involves tasks where both modalities (Audio, Image, Video) and task types (Captioning, QA) shift inconsistently.

To address the catastrophic forgetting inherent in this setting, the authors propose MoInCL, a framework that fine-tunes the LLM backbone using LoRA. MoInCL consists of two key components:
1. Pseudo Target Generation Module (PTGM): Mitigates task type shifts by leveraging the frozen pre-trained LLM to generate pseudo-QA pairs from captions.

2. Instruction-based Knowledge Distillation (IKD): Mitigates modality shifts by distilling the LLM's generative capabilities using a pure text instruction set.

**Audience:**

Yes

**Audience Explanation:**

The intersection of Multimodal Learning and Continual Learning is a high-interest area. The specific focus on "inconsistent" shifts reflects practical engineering challenges in deploying MLLMs. The paper offers a valuable benchmark that would interest researchers in both communities.

**Broader Impact Concerns:**

The authors adequately address efficiency and privacy benefits in their Broader Impact statement. No additional ethical concerns are raised by this work.

**Claims And Evidence:**

Yes

**Claims Explanation:**

The authors provide extensive experimental validation on the proposed MICL benchmark, covering six tasks and multiple orderings (including challenging orders in the Appendix). The ablation studies clearly isolate the contributions of the PTGM and IKD modules. However, the strength of the evidence is slightly dampened by the absence of state-of-the-art PEFT-CL baselines, meaning the superiority of the method is established against potentially weak methods.

**Requested Changes:**

1. Comparison with LoRA-based CL Methods (Critical)
The proposed method uses LoRA for fine-tuning the LLM. However, the baselines (EWC, LwF) are general regularization methods not optimized for Parameter-Efficient Fine-Tuning (PEFT). The paper effectively ignores the recent progress in PEFT-based Continual Learning. To validate the effectiveness of MoInCL, the authors should compare against (or at least extensively discuss) methods like O-LoRA [A] and its line of more recent works. If the proposed IKD and PTGM modules are truly necessary, they should outperform simply enforcing orthogonality in the LoRA ranks.

[A] Orthogonal Subspace Learning for Language Model Continual Learning

2. Clarification of "Memory-Free" Constraint (Strengthening)
The paper claims a "memory-free" setting but utilizes the "Natural Instructions" dataset for IKD. While this is not task data, it serves as a "replay buffer of generic linguistic capabilities."  Please explicitly categorize the role of the "Natural Instructions" dataset in the problem formulation. Is the assumption that a generic text corpus is always available? A brief sensitivity analysis on the size or domain of this dataset would strengthen the robustness of the IKD component.

3. Quality of Pseudo-Generated QA Pairs (Strengthening)
The PTGM relies on a three-round prompting strategy using the frozen LLM to generate QA pairs. Please provide a qualitative or quantitative assessment of these generated pairs. Does the frozen LLM hallucinate? If the visual domain is specialized, does the generic LLM fail to generate relevant questions? A small random sample analysis of $(\tilde{t}, \tilde{y})$ pairs would provide necessary confidence in this module.

4. Computational Overhead Analysis (Strengthening)
Section 4.4.3 mentions a ~40% increase in training time. Please clarify if this accounts for the online generation cost of PTGM. Explicitly stating the training throughput (samples/sec) compared to standard fine-tuning would be helpful for practitioners assessing the trade-off between anti-forgetting and efficiency.

5. Evaluation of General LLM Capabilities (Strengthening) While the paper introduces Instruction-based Knowledge Distillation (IKD) using the "Natural Instructions" dataset to preserve the LLM's generative abilities, the evaluation focuses exclusively on multimodal task metrics (CIDEr, Accuracy). It remains unclear if the continual fine-tuning comes at the cost of the backbone's general intelligence. Please provide an analysis (or discussion) comparing the model's performance on general domain benchmarks (e.g., MMLU, GSM8K, or general instruction following) before and after the continual learning process. Does the model retain its general reasoning and math capabilities, or does the adaptation to inconsistent modalities induce catastrophic forgetting in the fundamental LLM backbone?

---

> ### Author Response · Authors · 2026-02-27
> **Response to Reviewer VK5i**
>
> We sincerely thank the Reviewer VK5i for the constructive feedback! We have addressed your concerns below and revised our paper accordingly, and the updated parts are highlighted using blue color. If there are any additional questions, we are willing to address them and revise our paper.
>
> > ### **Q1: Comparison with LoRA-based CL Baselines. The baselines (EWC, LwF) are general regularization methods not optimized for Parameter-Efficient Fine-Tuning (PEFT).**
>
> Thank you very much for your suggestion.
>
> We would like to clarify that all compared baselines adopt exactly the same trainable-parameter configuration as our method for a fair comparison. For our method and all baselines, we fine-tune the LLM component with LoRA, freeze the modality encoders, and fully fine-tune the modality projection modules that map encoder features into the LLM input space. Therefore, EWC/LwF are trained and evaluated under the same PEFT setting as MoInCL.
>
> To further address your concern regarding recent PEFT-based continual learning approaches, we additionally compared with HiDe-LLaVA [1], which is one of the most recent works of PEFT (LoRA)-based continual learning methods for multimodal large language models.
>
> The comparison results are summarized below:
>
>
> Order 1:
> | Methods                        | Avg. CIDEr           $\uparrow$  | Avg. Acc.           $\uparrow$ | Avg. Forget.          $\downarrow$ |
> |--------------------------------|----------------------------------|--------------------------------|------------------------------------|
> | HiDe-LLaVA [1]                 | 25.27                            | 38.35                          | 46.47%                             |
> | **MoInCL**                     | **55.31**                        | **42.29**                      | **14.21%**                         |
>
>
> Order 2:
> | Methods                        | Avg. CIDEr           $\uparrow$  | Avg. Acc.           $\uparrow$ | Avg. Forget.          $\downarrow$ |
> |--------------------------------|----------------------------------|--------------------------------|------------------------------------|
> | HiDe-LLaVA [1]                 | 13.93                            | **46.15**                      | 46.18%                             |
> | **MoInCL**                     | **51.13**                        | 45.22                          | **8.93%**                          |
>
>
> These results demonstrate that simply enforcing orthogonality or structural decoupling in LoRA ranks is insufficient for our proposed modality-inconsistent continual learning. And the proposed IKD and PTGM modules are therefore necessary to effectively maintain transferable knowledge across tasks and modalities.
>
> The associated results have been included in the revised manuscript (Tables 1, 3, 4, 14, and 15).
>
> [1] HiDe-LLaVA: Hierarchical Decoupling for Continual Instruction Tuning of Multimodal Large Language Model. In *ACL* 2025.
>
> > ### **Q2: Clarifying the Memory-Free Assumption and Robustness of IKD to Instruction Corpus Variations.**
>
> Thank you for your suggestion. We have explicitly clarified the role of the Natural Instructions dataset in our memory-free setting in Section 3.1 by adding the following statement:
>
> *Please note that the term memory-free refers strictly to the absence of replay buffers that store multimodal samples from previous tasks. In our proposed MoInCL method, we introduce an auxiliary text-only instruction corpus solely for the IKD component (see Sec 3.4). This corpus is not part of the MICL task stream, contains no modality-specific data, and it serves only as a regularization resource within our proposed MoInCL and does not alter the underlying memory-free problem formulation.*
>
> In addition, we conduct an additional experiment on Order 1 to evaluate the sensitivity of IKD to the size of the Natural Instructions (NI) corpus.
>
> | Methods                        | Avg. CIDEr  $\uparrow$  | Avg. Acc.   $\uparrow$ | Avg. Forget.   $\downarrow$ |
> |------|-----------|-------------|------|
> | w/o IKD (0% NI)     | 53.33 | 40.69  | 17.82%                             |
> | IKD with 50% NI                | 55.04 | 42.17 | 14.96%                             |
> | IKD with 100% NI               | **55.31**                        | **42.29**                      | **14.21%**                         |
>
> The results show that even with a substantially reduced instruction corpus (50%), MoInCL maintains comparable performance and forgetting reduction. While removing IKD degrades performance and increases forgetting, the performance gap between 50% and 100% NI remains relatively small under the current corpus setting. This suggests that, IKD does not exhibit strong sensitivity to corpus size. We note that broader domain coverage or larger instruction diversity may potentially provide additional benefits, but our results indicate that IKD does not critically depend on the full corpus to be effective.
>
> We have included this analysis in Section 4.4.4.

---

> > ### Author Response · Authors · 2026-02-27
> > **Response to Reviewer VK5i (continued)**
> >
> > > ### **Q3: Qualitative Analysis of Generated Pseudo QA Pairs.**
> >
> > Thank you for the suggestion! We have included qualitative analysis of the generated pseudo QA pairs of the Image Captioning dataset (Flickr30k) used in our experiments. The qualitative results are presented in Appendix G. From these randomly sampled examples, we observe that the frozen LLM generally produces semantically consistent and visually grounded questions and answers. The generated questions align with the image captions and focus on relevant visual attributes such as object identity, actions, counting, and scene context.
> >
> > Since the pseudo QA pairs are generated conditioned on image captions, the generation is grounded in textual descriptions rather than unconstrained visual images, which reduces the likelihood of hallucinated or irrelevant content. In our sampled examples, we do not observe systematic hallucination or off-topic questions.
> >
> >
> > > ### **Q4: Regarding the Computational Overhead Analysis.**
> >
> > Thank you for the question! The reported ~40% increase in training time refers to the additional optimization cost introduced by IKD and PTGM during gradient updates. Importantly, this number does not include the generation cost of PTGM pseudo-labels.
> >
> > Specifically, both pseudo captions and pseudo QA pairs are generated using frozen models (*i.e.*, the previous task trained model or frozen LLM). These pseudo samples can be pre-generated offline before training each new task. Therefore, they do not introduce additional per-epoch or per-iteration generation overhead during continual learning. In our implementation, all pseudo data are generated once at the beginning of each task and then reused throughout training of each task.
> >
> > We have clarified this in Sec. 4.4.3.
> >
> > > ### **Q5: Evaluation of General LLM Capabilities.**
> >
> >
> > Thank you for your valuable suggestion.
> >
> > To evaluate whether continual learning under modality inconsistency affects the general reasoning capabilities of the LLM backbone, we conduct zero-shot evaluation on the MMLU benchmark before and after each continual learning step on Order 1 with our proposed MoInCL method.
> >
> >
> > | Continual Learning Steps  | 0 (before continual learning) | 1      | 2      | 3      | 4      | 5      | 6      |
> > |---------------------------|-------------------------------|--------|--------|--------|--------|--------|--------|
> > | Accuracy (%)              | 43.87                         | 43.27  | 43.01. | 43.41  | 43.42  | 43.13  | 43.34  |
> >
> >
> > As shown above, the backbone's MMLU accuracy remains highly stable throughout the entire continual learning process, with fluctuations within $\pm$1\% of the initial performance. We do not observe catastrophic forgetting in the LLM's general reasoning capability. These results indicate that continual adaptation under MoInCL does not degrade the fundamental reasoning and knowledge capacity of the LLM backbone. The LoRA-based adaptation together with IKD preserves general reasoning capability while enabling effective multimodal continual learning.
> >
> > We have included these results in Sec. 4.4.5.

---

### Review · Reviewer_u5zb · 2026-02-21

**Summary Of Contributions:**

This paper introduces Modality-Inconsistent Continual Learning (MICL), a continual learning setting for multimodal large language models (MLLMs) where tasks arrive sequentially with both modality shifts (image, audio, video) and task-type shifts (captioning vs. question answering), which together exacerbate catastrophic forgetting. To tackle this, the authors propose MoInCL, combining a Pseudo Targets Generation Module (PTGM) to reduce forgetting from task-type changes within previously seen modalities, and Instruction-based Knowledge Distillation (IKD) to preserve modality- and task-aware knowledge when new modalities are introduced. They benchmark MICL on six incremental tasks (Image/Audio/Video × Captioning/QA) and show that MoInCL consistently and significantly outperforms representative and state-of-the-art continual learning baselines, effectively mitigating forgetting from both modality and task-type perspectives.

Strengths:

(1) The proposed modality-inconsistent, task-incremental continual learning setting is timely and compelling. It better reflects real-world MLLM deployment, and it has clear potential to scale to broader modality and task mixes, making it a valuable testbed for building stronger general-purpose MLLMs.

(2) The PTGM design is lightweight and largely relies on prompt engineering, yet it is empirically effective. MoInCL shows strong and robust performance across different task orders, suggesting the approach is practical and not overly sensitive to the chosen continual learning curriculum.

**Audience:**

Yes

**Audience Explanation:**

As noted above, the setting proposed in this paper is worth exploring for the continual learning community. While the method itself is somewhat incremental with existing techniques and can have limitations in some cases (e.g., LLM hallucination when synthesising QA pairs), it could serve as a useful baseline for future work, especially if the authors release the code and the benchmark setting to support follow-up research.

**Claims And Evidence:**

Yes

**Claims Explanation:**

Yes and No.
Overall, the setting proposed in this paper is worth exploring, and the proposed method shows strong empirical performance, outperforming competitive baselines. The authors also support the value of the two key modules, PTGM and IKD, through ablation studies across multiple task orders and experimental settings. However, the experimental part can be more comprehensive given my suggestions above (e.g., more baselines, more advanced models).

**Requested Changes:**

## Major Weaknesses

i)	On the design of PTGM:

- The PTGM builds pseudo-QA pairs from the text context $M$ and target $y$ only, without using the actual image, audio, or video evidence. For QA, where the answer often depends on what is in the modality input, I worry these text-only pseudo targets could push the model to lean more on language priors and pay less attention to the multimodal signal, which is already a common failure mode for many VLMs and MLLMs. Could the authors explain why this still gives useful supervision and does not encourage shortcut learning?

- In Section 3.3, the paper states that only $x$ is used for generating pseudo targets,'' but $x$ does not appear to be used in the PTGM description. Please clarify this.
-	PTGM resembles synthesising pseudo data for distillation rather than storing original data. If pseudo data quality is poor due to LLM hallucination, it may introduce noise and harm performance. Please discuss how pseudo target quality is controlled and how sensitive performance is to generation errors.

ii)	On the motivation of IKD:

- The paper states that different modalities do not share the modality encoder or modality projection (e.g., visual encoder). However, forgetting can still occur in shared components such as attention layers or PEFT modules that have been updated to support new modalities. Preserving only the LLM side may therefore be insufficient.
-	If the instruction set is text-only and comes from a similar domain to the earlier tasks, then the smaller amount of forgetting might just be because the data are familiar, not because the model is really retaining modality-specific ability. Could you clarify this, and ideally add a control to separate domain overlap from true multimodal retention?

iii)	On the novelty of method design:

-	The core ingredients appear to be pseudo data generation or external instruction data, plus distillation between old and new models. The methodology seems incremental, with limited new algorithmic design beyond combining known techniques.

iv)	On the experimental designs/results:

- Please include additional continual learning baselines, such as LAE (ICCV 2023) [1], HiDe-Prompt (NeurIPS 2023) [2], ProgPrompt (ICLR 2023) [3], and Bisecle (NeurIPS 2025) [4], for a more complete comparison.

[1] Gao, Q., Zhao, C., Sun, Y., Xi, T., Zhang, G., Ghanem, B. and Zhang, J., 2023. A unified continual learning framework with general parameter-efficient tuning. In Proceedings of the IEEE/CVF International Conference on Computer Vision (pp. 11483-11493).

[2] Wang, L., Xie, J., Zhang, X., Huang, M., Su, H. and Zhu, J., 2023. Hierarchical decomposition of prompt-based continual learning: Rethinking obscured sub-optimality. Advances in Neural Information Processing Systems, 36, pp.69054-69076.

[3] Razdaibiedina, A., Mao, Y., Hou, R., Khabsa, M., Lewis, M. and Almahairi, A., Progressive Prompts: Continual Learning for Language Models. In The Eleventh International Conference on Learning Representations.

[4] Tan, Y., Hu, X., Xue, H., de Melo, C.M. and Salim, F.D., Bisecle: Binding and Separation in Continual Learning for Video Language Understanding. In The Thirty-ninth Annual Conference on Neural Information Processing Systems.

-	Table 3 shows the largest forgetting reduction on Image Captioning. Please provide intuition for why this task benefits most. The table also suggests positive transfer (negative forgetting) on ImageQA. Please explain this behaviour and identify which component drives it.
-	I am interested in the ImageQA to VideoQA transition discussed in Section 4.4.2. Could the authors include experiments on this setting? My intuition is that learning ImageQA, which does not require temporal reasoning, may lead to negative transfer when moving to VideoQA, so I would like to see whether this happens in your results.
-	To strengthen both coverage and reproducibility, I suggest testing more MLLMs (e.g., Qwen-VL and LLaVA), running ablations across different LLaMA model sizes, and releasing the source code to support community follow-up work.

## Minor Weaknesses:

- Equation (2) is not presented clearly. Please state the constraints explicitly and clarify the set membership and notation. For instance, the text mentions conditions like $M \in L_M$, $P_i \notin L^T_M$, and the index $i$ but these do not appear in the equation itself, so it is unclear how they are enforced.
-	The paper should include concrete examples of the pseudo target $\tilde{y}$ (in Appendix A) to clearly distinguish $y$, $\tilde{y}$, and any other variants such as $\bar{y}$. The current demonstration without examples is hard to understand.

---

> ### Author Response · Authors · 2026-02-27
> **Response to Reviewer u5zb**
>
> We sincerely thank the Reviewer u5zb for the constructive feedback! We have addressed your concerns below and revised our paper accordingly, and the updated parts are highlighted using blue color. If there are any additional questions, we are willing to address them and revise our paper.
>
> > ### **Q1: Risk of language shortcut learning of pseudo QA pairs.**
>
> We agree that shortcut learning via language priors is a common issue in multimodal models. However, we clarify that the pseudo QA pairs in PTGM are not generated independently of the modality input. Instead, they are derived from the ground-truth target caption $y$, which itself is a semantic description of the modality input $x$ (image, video, or audio). Therefore, the pseudo QA supervision remains grounded in the multimodal signal through this semantic chain: $x \rightarrow y \rightarrow$ Pseudo QA.
>
> Importantly, during training, the model is still conditioned on the original multimodal input $x$ when optimizing the pseudo QA objective. The pseudo QA loss is computed under the full multimodal input rather than text-only input. As a result, the model must attend to modality features to produce correct answers, rather than relying solely on language priors.
>
> Empirically, if PTGM encouraged shortcut learning, we would expect degraded multimodal performance. However, as shown in our experiments, PTGM consistently improves multimodal metrics (*e.g.*, CIDEr and QA accuracy), suggesting that it reinforces modality-grounded representations rather than weakening them.
>
> > ### **Q2: Clarification on the Use of $\boldsymbol{x}$ and $\boldsymbol{y}$ in PTGM.**
>
> Thank you for pointing this out. We agree that the original wording in Sec. 3.3 was unclear due to a typographical error.
>
> The sentence "Please note that only $\boldsymbol{x}$ is used for generating pseudo targets, while only $\boldsymbol{y}$ is utilized for generating pseudo QA pairs." should be revised to: "Please note that only $\boldsymbol{x}$ is used for generating pseudo captions, while only $\boldsymbol{y}$ is utilized for generating pseudo QA pairs."
>
> In PTGM, two generation branches are involved: pseudo captions are generated from the modality input $\boldsymbol{x}$ (*e.g.*, when training on a QA task), whereas pseudo QA pairs are generated from the ground-truth caption $\boldsymbol{y}$ (*e.g.*, when training on a captioning task). We have corrected this wording in the revised manuscript to avoid ambiguity.
>
> > ### **Q3: On Pseudo-Target Quality Control and Sensitivity to Generation Errors in PTGM.**
>
> Thank you for the insightful comment.
>
> We would like to clarify that both pseudo captions and pseudo QA pairs are introduced to mitigate task-type shift in PTGM, but they rely on different supervision mechanisms and thus differ in their sensitivity to generation imperfections:
>
> **Pseudo Captions.**
> Pseudo captions are generated by the frozen previous-task model and are used to retain and preserve the learned captioning ability of the old model. The objective of pseudo caption generation is not to approximate ground-truth captions, but to maintain the old model's output behavior through distillation. Therefore, even if the generated captions are imperfect, they faithfully reflect the old model's learned knowledge and captioning abilities. The goal of this branch is behavioral preservation rather than introducing new supervision, making it inherently robust to moderate generation errors.
>
>
> **Pseudo QA Pairs.**
> We acknowledge that pseudo QA pairs generated by a frozen LLM may contain noise or incomplete semantic coverage. However, they are generated conditioned on the ground-truth caption, which provides a semantically grounded textual description of the modality input. This grounding reduces unconstrained hallucination. Moreover, we have discussed the potential limitations of pseudo QA generation and its imperfect semantic coverage in the Limitations section, where we explicitly acknowledge this trade-off.
>
> Empirically, if pseudo-target noise were severely detrimental, we would expect PTGM to degrade continual performance. However, our ablation results show that introducing PTGM consistently improves multimodal metrics and reduces forgetting compared to removing PTGM. This indicates that, in practice, the framework is robust to moderate generation imperfections and that the pseudo supervision provides beneficial regularization rather than harmful noise.

---

> > ### Author Response · Authors · 2026-02-27
> > **Response to Reviewer u5zb (continued)**
> >
> > > ### **Q4: On the Sufficiency of Preserving the Shared LLM Component for Modality Shift.**
> >
> > Thank you for this important comment.
> >
> > We would like to clarify that in the model architecture, different modalities do not share modality encoders or modality projection modules. Each modality has its own dedicated encoder and projector. The only shared component across modalities is the LLM backbone, including its attention layers and LoRA-based PEFT modules. Therefore, any cross-modality interference or forgetting can only occur in the shared LLM component. Since encoders and projectors are modality-specific and not reused across tasks, preserving the LLM side is both necessary and sufficient for mitigating modality shift in our setting.
> >
> > > ### **Q5: On Domain Overlap and True Multimodal Retention.**
> >
> > Thank you for this thoughtful comment.
> >
> > We would like to clarify that the instruction set used in IKD is a pure text corpus and does not contain any modality-specific inputs (*e.g.*, image, video, or audio). In contrast, each continual task in MICL is trained on modality-specific datasets, where supervision depends on multimodal inputs. Therefore, the instruction corpus does not overlap with the modality-specific data distributions of the continual tasks.
> >
> > While both IKD and task prompts may adopt an instruction-style format, there is no overlap in modality semantics. The instruction corpus does not contain visual, audio, or video grounding, and therefore cannot directly provide modality-specific familiarity. Moreover, IKD regularizes the shared LLM representations that are reused across different modalities. Since modality shift arises from updating this shared backbone when adapting to new modality-conditioned tasks, stabilizing the shared LLM helps mitigate cross-modality interference rather than exploiting any domain overlap in the instruction corpus.
> >
> > We have clarified this point in Sec. 3.4 in the revised manuscript.
> >
> > > ### **Q6: On the Novelty of Method Design.**
> >
> > Thank you for the concern.
> >
> > We would like to clarify that our contribution is not a simple combination of existing techniques, but is driven by a new continual learning setting that introduces challenges not addressed in prior works.
> >
> > Specifically, in our proposed Modality-Inconsistent Continual Learning (MICL) scenario, tasks simultaneously involve modality shift and task-type shift. Unlike conventional continual learning, which typically assumes a fixed task format or shared modality space, MICL requires the model to continually adapt across heterogeneous modalities (image, video, audio) and heterogeneous task types (captioning vs. QA). This joint inconsistency introduces a new form of structural interference in MLLMs that existing CL methods are not designed to handle.
> >
> > Our technical contributions are therefore scenario-driven:
> >
> > - **PTGM** is designed to explicitly counteract task-type shift by synthesizing cross-task-type pseudo targets that preserves modality-conditioned behaviors across task type boundaries.
> >
> > - **IKD** addresses modality-induced interference by stabilizing the shared LLM backbone under modality inconsistency, preventing representational drift in the shared components.
> >
> > Importantly, these components are not generic distillation or pseudo-data strategies applied in isolation. They are jointly formulated to target the two orthogonal yet interacting failure modes unique to MICL. Without modeling this interaction, existing continual learning approaches are insufficient.
> >
> > Thus, the novelty of our work lies in (1) introducing a previously unaddressed continual learning setting for MLLMs and (2) designing a unified, technically grounded framework that directly resolves its structural challenges.

---

> ### Author Response · Authors · 2026-02-27
> **Response to Reviewer u5zb (continued)**
>
> > ### **Q7: Additional Baseline.**
>
> Thank you for the suggestion.
>
> We further include experimental comparisons with HiDe-LLaVA [1], one of the most recent continual learning methods for MLLMs. The results are summarized below.
>
> Order 1:
> | Methods                        | Avg. CIDEr           $\uparrow$  | Avg. Acc.           $\uparrow$ | Avg. Forget.          $\downarrow$ |
> |--------------------------------|----------------------------------|--------------------------------|------------------------------------|
> | HiDe-LLaVA [1]                 | 25.27                            | 38.35                          | 46.47%                             |
> | **MoInCL**                     | **55.31**                        | **42.29**                      | **14.21%**                         |
>
>
> Order 2:
> | Methods                        | Avg. CIDEr           $\uparrow$  | Avg. Acc.           $\uparrow$ | Avg. Forget.          $\downarrow$ |
> |--------------------------------|----------------------------------|--------------------------------|------------------------------------|
> | HiDe-LLaVA [1]                 | 13.93                            | **46.15**                      | 46.18%                             |
> | **MoInCL**                     | **51.13**                        | 45.22                          | **8.93%**                          |
>
>
> MoInCL consistently achieves significantly lower forgetting and substantially stronger generation performance across task orders, demonstrating its effectiveness compared with recent multimodal CL approaches. We have incorporated this new comparison in our revised manuscript (Tables 1, 3, 4, 14, and 15).
>
>
> [1] HiDe-LLaVA: Hierarchical Decoupling for Continual Instruction Tuning of Multimodal Large Language Model. In *ACL* 2025.
>
> > ### **Q8: On the Larger Forgetting Reduction in Image Captioning and the Positive Transfer on Image QA in Tab. 3 (Order 1).**
>
> **Image Captioning.**
> In Order 1, Image Captioning is learned relatively early and is subsequently followed by three consecutive QA tasks (including Image QA), resulting in a prolonged shift from captioning-style generation to QA-style prediction, in addition to modality shifts. And the final Image QA task reuses the image modality and partially reduces modality discrepancy, the sustained task-type shift remains the dominant source of interference for captioning. Among these factors, task-type shift is particularly disruptive to captioning, as later QA-oriented training shifts the generative behavior of the shared LLM backbone and, in the case of Image QA, also adapts the image-specific modality projector, both of which can interfere with previously learned captioning behaviors. Since PTGM explicitly preserves captioning-style behaviors through cross-task-type supervision and IKD stabilizes the shared LLM representation, Image Captioning — being highly sensitive to changes in the model's generative distribution — benefits most visibly from these mechanisms.
>
> **Image QA.**
> In Order 1, Image QA is followed by Video Captioning. During Video Captioning training, PTGM generates pseudo QA pairs conditioned on caption targets, thereby reintroducing QA-style supervision even when the primary task objective is captioning. This mechanism helps maintain and even reinforce QA behaviors in the shared LLM, which can result in mild positive transfer on the previously learned Image QA task. Moreover, although modality shift (image $\rightarrow$ video) occurs, both tasks operate within the visual domain. Stabilization of the shared LLM through IKD further reduces cross-modality interference, contributing to the observed behavior.
>
> We have included this analysis in Sec. 4.2 in our revised manuscript.
>
> > ### **Q9: Task Transfer Effectiveness from Image QA to Video QA.**
>
> Thank you for the insightful suggestion.
>
> We conducted additional experiments to evaluate the transfer effect from Image QA to Video QA. The results are summarized below:
>
>
> | Video QA       | Image QA $\rightarrow$ Video QA  |
> |----------------|---------------------------------|
> | 44.81          | 45.74                           |
>
>
> We observe that initializing Video QA learning after Image QA leads to improved performance (45.74 vs. 44.81), indicating positive rather than negative transfer.
>
> Although Image QA does not require temporal reasoning, it trains the shared LLM to perform spatial reasoning and QA behaviors grounded in visual inputs. These capabilities remain beneficial when adapting to Video QA, where temporal reasoning is required in addition to spatial reasoning. The shared LLM backbone can therefore reuse previously acquired QA-related representations while incorporating temporal information from the video modality.
>
> We have included these results in Tab. 6 in our revised manuscript.

---

> ### Author Response · Authors · 2026-02-27
> **Response to Reviewer u5zb (continued)**
>
> > ### **Q10: On Evaluating Additional MLLMs and Larger Model.**
>
> Regarding evaluating additional MLLMs such as Qwen-VL and LLaVA, we note that these models are fully pretrained multimodal large language models trained on large-scale vision-language corpora with joint multimodal objectives. In contrast, our MICL setting is designed to study modality-inconsistent continual adaptation starting from a pretrained LLM backbone, without assuming a fully pretrained multimodal model. Using such pretrained MLLMs would substantially change the experimental setting and introduce strong prior multimodal capabilities that are not aligned with the intended problem formulation. Furthermore, models such as Qwen-VL and LLaVA are primarily designed for visual modalities and do not support audio tasks, whereas MICL explicitly involves image, video, and audio tasks. Therefore, they are not structurally compatible with our full task setting.
>
>
> To evaluate scalability on larger MLLMs, we conducted additional experiments using LLaMA-3.1-8B-Instruct as the LLM component (instead of LLaMA-3.2-1B-Instruct used in the main experiments). The experimental results are presented in the following table and clearly show that our method consistently outperforms the baselines, demonstrating the effectiveness and robustness of our method when scaling the LLM size from 1B to 8B parameters.
>
>
> | Methods                        | Avg. CIDEr           $\uparrow$  | Avg. Acc.           $\uparrow$ | Avg. Forget.          $\downarrow$ |
> |--------------------------------|----------------------------------|--------------------------------|------------------------------------|
> | Fine-tuning                    | 30.42                            | 50.07                          | 39.02%                             |
> | LwF                            | 33.15                            | 50.31                          | 38.46%                             |
> | EWC                            | 36.88                            | 46.82                          | 37.37%                             |
> | EWF                            | 28.65                            | 50.92                          | 37.47%                             |
> | PathWeave                      | 35.72                            | 46.62                          | 40.55%                             |
> | **MoInCL**                     | **65.24**                        | **52.94**                      | **9.21%**                          |
>
> We have included these results in Appendix E in our revised manuscript.
>
>
> > ### **Q11: Releasing Source Code.**
>
> Thank you for the suggestion. We will release our code to ensure full reproducibility upon acceptance
>
>
> > ### **Q12: Explicitly State the Constraints of Eq. (2).**
>
> Thank you for the suggestion. We have included an explicit explanation of the constraints under Eq. (2) in the revised manuscript:
>
> For task $\mathcal{T}\_{i}$ with modality $M_i$ and task type $P_i$, if $M_i$ is a previously seen modality (*i.e.*, $M_i\in LM$), and if the current task type $P_i$ is not a learned task type of the current modality $M_i$ (*i.e.*, $P_i\notin LT_{M_i}$), we generate pseudo targets via $PTGM$ for the learned task type $p$ of modality $M_i$.
>
>
> > ### **Q13: Include Concrete Examples of the Pseudo Target in Appendix A.**
>
> Thank you for your suggestion.
>
> We have included concrete examples of the generated pseudo targets through the described process in Appendix A.
>
> These examples are presented in Appendix G (Fig. 9), and we have added an explicit cross-reference to Fig. 9 in Appendix A to guide readers.

---

> > ### Comment · Reviewer_u5zb · 2026-03-13
> >
> > Thank you for your responses and clarifications.
> >
> > Regarding shortcut learning, pseudo-target quality control, and sensitivity to generation errors in PTGM, I agree with your explanation that the pseudo-QA is generated from the ground-truth target caption, which helps minimise these issues. This design now makes sense to me. However, the quality of the generated pseudo-QA is still constrained by the limited information in the original caption and by the capability of the previous model. Looking beyond the current setting, how could the pseudo-QA be further enriched while still ensuring its quality and trustworthiness?
> >
> > The authors have addressed my concern regarding the novelty of the method design. While the core techniques have been explored in prior work, the way they are extended and adapted to the proposed setting is meaningful and could encourage follow-up research.
> >
> > I also appreciate the additional results. Overall, I quite like the idea of pseudo replay in MoInCL. My remaining question concerns a more general continual task learning setting in which generative tasks appear. In such cases, how can pseudo data be generated to rehearse and preserve the model's generation capability?

---

> > > ### Author Response · Authors · 2026-03-15
> > > **Response to Reviewer u5zb**
> > >
> > > Thank you for the thoughtful follow-up question and for the positive feedback on the design of MoInCL.
> > >
> > >
> > > > ### **Q1: On the richness and reliability of pseudo QA generation.**
> > >
> > > Thank you for the question. As the reviewer noted, the quality of pseudo QA pairs is ultimately bounded by the information contained in the ground-truth caption and the capability of the previous model used to generate pseudo captions. In our setting, we assume that the captioning datasets used for training provide sufficiently informative descriptions of the underlying image/video/audio content. Under this common assumption in captioning benchmarks, the ground-truth captions typically contain the key semantic elements of the input, which provide a reliable basis for generating meaningful pseudo-QA pairs. If a caption lacks sufficient information to support such generation, it would also limit the effectiveness of the captioning task itself.
> > >
> > > Moreover, pseudo-QA richness could be further improved through several directions while maintaining reliability: for example, leveraging multiple QA generations per caption, applying consistency filtering across multiple LLM samples, or incorporating additional contextual signals (*e.g.*, structured prompts). Since PTGM generates pseudo QA pairs before the start offline, such quality control mechanisms can be integrated without affecting training stability.
> > >
> > >
> > > > ### **Q2: On pseudo data generation for more general generative tasks.**
> > >
> > > Thank you for the question. In the setting of our proposed MICL scenario, we focus on continual learning for MLLMs where the model output space is text. This is consistent with the common formulation of LLMs and MLLMs, which typically generate textual outputs even when processing multimodal inputs. Under this formulation, tasks such as captioning and question answering share a unified text generation interface, which allows pseudo replay to be implemented through pseudo captions and pseudo QA pairs.
> > >
> > > Extending the framework to settings where the output modality itself is generative (*e.g.*, image, video, or audio generation) would require pseudo data generation mechanisms tailored to those modalities, typically involving diffusion or autoregressive generation backbones. Such scenarios are conceptually related but fall outside the scope of our current MLLM continual learning formulation. We view this as an interesting direction for future work, where modality-specific generative replay mechanisms could be explored.

---

### Decision · Action_Editor_uZkw · 2026-04-09

**Recommendation:** Accept as is

**Audience:**

Yes

**Audience Explanation:**

Yes. The paper introduces a novel and practically relevant continual learning setting for multimodal large language models, combining both modality and task-type shifts. This problem formulation and the accompanying empirical study are likely to be of interest to researchers in continual learning, multimodal learning, and foundation models.

**Claims And Evidence:**

Yes

**Claims Explanation:**

Yes. In particular, the revised manuscript added comparisons with recent baselines, larger-scale experiments, and additional evaluations showing that the model preserves general reasoning ability during continual learning. The rebuttal also addressed earlier concerns about pseudo-target generation, hallucination, and efficiency.